# Different Requirements of CBFB and RUNX2 in Skeletal Development among Calvaria, Limbs, Vertebrae and Ribs

**DOI:** 10.3390/ijms232113299

**Published:** 2022-10-31

**Authors:** Qing Jiang, Xin Qin, Kenichi Nagano, Hisato Komori, Yuki Matsuo, Ichiro Taniuchi, Kosei Ito, Toshihisa Komori

**Affiliations:** 1Department of Molecular Bone Biology, Nagasaki University Graduate School of Biomedical Sciences, Nagasaki 852-8588, Japan; 2Institute of Orthopaedics, Suzhou Medical College, Soochow University, Suzhou 215006, China; 3Department of Oral Pathology and Bone Metabolism, Nagasaki University Graduate School of Biomedical Sciences, Nagasaki 852-8588, Japan; 4Laboratory for Transcriptional Regulation, RIKEN Research Center for Allergy and Immunology, Yokohama 230-0045, Japan

**Keywords:** RUNX1, RUNX2, RUNX3, CBFB, cleidocranial dysplasia, calvaria, limb bone, vertebra, rib, osteoblast

## Abstract

RUNX proteins, such as RUNX2, regulate the proliferation and differentiation of chondrocytes and osteoblasts. Haploinsufficiency of *RUNX2* causes cleidocranial dysplasia, but a detailed analysis of *Runx2*^+/−^ mice has not been reported. Furthermore, CBFB is required for the stability and DNA binding of RUNX family proteins. CBFB has two isoforms, and CBFB2 plays a major role in skeletal development. The calvaria, femurs, vertebrae and ribs in *Cbfb2*^−/−^ mice were analyzed after birth, and compared with those in *Runx2*^+/−^ mice. Calvarial development was impaired in *Runx2*^+/−^ mice but mildly delayed in *Cbfb2*^−/−^ mice. In femurs, the cortical bone but not trabecular bone was reduced in *Cbfb2*^−/−^ mice, whereas both the trabecular and cortical bone were reduced in *Runx2*^+/−^ mice. The trabecular bone in vertebrae increased in *Cbfb2*^−/−^ mice but not in *Runx2*^+/−^ mice. Rib development was impaired in *Cbfb2*^−/−^ mice but not in *Runx2*^+/−^ mice. These differences were likely caused by differences in the indispensability of CBFB and RUNX2, the balance of bone formation and resorption, or the number and maturation stage of osteoblasts. Thus, different amounts of CBFB and RUNX2 were required among the bone tissues for proper bone development and maintenance.

## 1. Introduction

RUNX2 belongs to RUNX family transcription factors composed of RUNX1, RUNX2 and RUNX3, and is required for the commitment of multipotent mesenchymal cells to an osteoblast lineage of cells, the proliferation of osteoblast progenitor cells and the expression of major bone matrix protein genes, including *Col1a1*, *Col1a2*, *Spp1*, *Ibsp* and *Bglap*&*Bglap2*, a transcription factor *Sp7* and a protease *Htra1* [1,2]. Additionally, RUNX1 and RUNX3 induce bone formation [3,4]. RUNX2 and RUNX1 induce chondrocyte proliferation and maturation, and RUNX3 is partly involved in chondrocyte maturation [5,6]. Furthermore, RUNX2 enhances bone resorption by inducing *Tnfsf11* expression in osteoblasts [7,8]. Notably, Haploinsufficiency of *RUNX2* causes cleidocranial dysplasia, which is characterized by open fontanelles and sutures, hypoplastic clavicles, supernumerary teeth and short stature [9,10,11].

CBFB is a cotranscription factor, which is ubiquitously expressed. CBFB heterodimerizes with RUNX family transcription factors, and enhances their DNA binding capacity [12,13]. *Cbfb*-deficient (*Cbfb*^−/−^) mice die between embryonic days 11.5–13.5 due to the absence of definitive hematopoiesis in the liver, similarly to *Runx1*^−/−^ mice, indicating that RUNX1 and CBFB are essential for definitive hematopoiesis [14,15,16]. The knock-in of green fluorescent protein (GFP) into the coding region of *Cbfb* maintained sufficient function in hematopoietic cells to bypass the early embryonic lethality [17]. Furthermore, the introduction of *Cbfb* into *Cbfb*^−/−^ mice using *Cbfb* transgenic mice under the control of *Tek1* or *Gata1* promoter, which directs *Cbfb* to hematopoietic progenitor cells, rescued definitive hematopoiesis in *Cbfb*^−/−^ mice [18,19]. These mice survived until birth, but showed severely impaired intramembranous and endochondral ossification. Although the phenotypes were milder than those in *Runx2*^−/−^ mice, the osteoblast differentiation and chondrocyte maturation were severely impaired, indicating that CBFB is also required for RUNX2-dependent bone development [17,18,19]. Moreover, the functions of CBFB in osteoblast differentiation and chondrocyte maturation were confirmed by generating *Cbfb* conditional knockout mice using *Sp7*-Cre, *Col2a1*-Cre or *Prrx1*-Cre transgenic mice and *Dermo1*-Cre knock-in mice [20,21,22,23,24,25]. CBFB regulates the functions of RUNX family proteins not only by enhancing their DNA binding capacity, but also by stabilizing the proteins [23,24]. 

CBFB has two functional isoforms, CBFB1 and CBFB2, which are formed by alternative splicing using donor splicing sites located inside exon 5 and at the 3′ terminus of exon 5, respectively and an acceptor splice site located at the 5′ terminus of exon 6 [12,26]. *Cbfb2* expression is about three times higher than *Cbfb1* expression in many tissues, including the calvaria, limbs, liver, thymus and brain [26]. *Cbfb1*-specific deletion by mutating the donor splicing signal sequence showed normal skeletal development, whereas *Cbfb2*-specific deletion by mutating the donor splicing signal sequence showed impaired endochondral and intramembranous ossification due to the delay in chondrocyte maturation and osteoblast differentiation during embryogenesis. *Cbfb2* mRNA was upregulated in *Cbfb1*^−/−^ mice, and the level of *Cbfb* was similar to that in wild-type mice, whereas *Cbfb1* mRNA was reduced in *Cbfb2*^−/−^ mice and the level of *Cbfb* was one-fourth of that in wild-type mice. However, CBFB1 was more potent than CBFB2 in chondrocyte and osteoblast differentiation and in DNA binding with RUNX2 in vitro [26]. The protein levels of RUNX1, RUNX2 and RUNX3 in *Cbfb1*^−/−^ mice were similar to those in wild-type mice, whereas those in *Cbfb2*^−/−^ mice were reduced compared with those in wild-type mice, although the levels of the reduction were different among RUNX family proteins and tissues [26]. 

Although open fontanelles and sutures and hypoplastic clavicles are observed in *Runx2*^+/−^ mice [10], the development of the other bones have not been evaluated. Furthermore, *Cbfb* deletion in the skeletal progenitors is lethal at the newborn stage [24], and the requirement of CBFB in skeletal development after birth remains to be investigated. The expression levels of *Cbfb* mRNA in *Cbfb2*^−/−^ mice were about one-fifth of that in wild-type mice, and half of the *Cbfb2*^−/−^ mice survived beyond 4 weeks of age. Here, we show the differential requirement of CBFB and RUNX2 in the bone development of calvaria, limbs, vertebrae and ribs after birth by comparing *Cbfb2*^−/−^ and *Runx2*^+/−^ mice.

## 2. Results

### 2.1. Protein Levels of RUNX1, RUNX2, RUNX3 and CBFB in Cbfb2^−/−^ and Runx2^+/−^ Mice 

The protein levels of RUNX1, RUNX2, RUNX3 and CBFB were examined using calvaria, trabecular bone from tibiae and femurs, ribs and vertebrae at 4 weeks of age (Figure 1). The reduction of RUNX1 in the *Cbfb2*^−/−^ mice was mild, except in the ribs. In the ribs, all RUNX family and CBFB proteins were more severely reduced in the *Cbfb2*^−/−^ mice with severe rib deformity than those with mild rib deformity, compared with those in wild-type mice (Figure 1F,G). The reduction of RUNX2 protein in the calvaria and trabecular bone of *Cbfb2*^−/−^ mice were 30 and 46%, respectively (Figure 1A,B,D,E), while that in the ribs was more severe, 75% (Figure 1F,G). In contrast, the level of RUNX2 protein in the vertebrae of *Cbfb2*^−/−^ mice was comparable with that of wild-type mice (Figure 1H,I). Among the RUNX family, the protein level of RUNX3 was most affected by the absence of CBFB2 and the reduction of RUNX3 protein in *Cbfb2*^−/−^ mice was 58–81% in the tissues (Figure 1). The reduction of CBFB protein in the *Cbfb2*^−/−^ mice was constant among the tissues, 76–85% (Figure 1). In calvaria, the protein level of RUNX2 in the *Runx2*^+/−^ mice was half of that in wild-type mice, and those of RUNX1, RUNX3 and CBFB were comparable with those in wild-type mice (Figure 1A,C). 

### 2.2. Skeletal Development of Cbfb1^−/−^, Cbfb2^−/−^ and Runx2^+/−^ Newborns

*Cbfb1*^−/−^ mice showed normal skeletal development in the newborn stage (Figure 2B,E,H,K,N,Q), and the trabecular bone volume, trabecular thickness, trabecular number, cortical area and cortical thickness were similar to those in wild-type mice at 10 weeks of age according to micro-CT analysis (Appendix A). Mineralization of the interparietal bone, supraoccipital bone (Figure 2A,C), ribs (Figure 2G,I), sternum (Figure 2D,F,G,I), vertebrae (Figure 2J,L), scapulae (Figure 2M,O) and pelvic bones (Figure 2P,R) was mildly delayed, and clavicles were thinner (Figure 2D,F) in *Cbfb2*^−/−^ newborns than those in wild-type newborns. In the scapulae in *Cbfb2*^−/−^ newborns, the whole area, mineralized area and the ratios of mineralized area in the whole area were lower than those in wild-type newborns (Figure 2S). Mineralized area in the calvaria was apparently reduced (Figure 3A,B), clavicles were severely hypoplastic (Figure 3C,D) and mineralization of sternum (Figure 3C–F), scapulae (Figure 3I,J) and pelvic bones (Figure 3K,L) was mildly delayed in *Runx2*^+/−^ newborns than those in wild-type newborns. In the scapulae in *Runx2*^+/−^ newborns, the whole area, mineralized area and the ratios of mineralized area in the whole area were lower than those in wild-type newborns (Figure 3M). There were no apparent differences in mineralization of ribs and vertebrae between wild-type and *Runx2*^+/−^ newborns (Figure 3E–H). The same phenotypes were confirmed in all newborns analyzed.

In the crossing of *Cbfb2*^+/−^ mice, the frequency of *Cbfb2*^−/−^ mice in the offspring during the embryonic stage was 24% (Appendix A). However, it was slightly reduced to 21% between postnatal day 0 (P0) and P2 and reduced further to 11% at 4 weeks of age. Thus, about half of the *Cbfb2*^−/−^ mice were estimated to have died by 4 weeks of age.

### 2.3. Delayed Closure of Posterior Frontal Suture in Cbfb2^−/−^ Mice

The processes of the closure of posterior frontal suture are unique and include the processes of endochondral ossification [27,28,29]. Since its closure was completely interrupted in *Runx2*^+/−^ mice [27], the processes were examined in *Cbfb2*^−/−^ mice at four different ages (Figure 4, Figure 5, Figure 6 and Figure 7). In wild-type mice, the frontal suture was open in the image of micro-CT at P7 (Figure 4A,B). In the histological analysis of wild-type mice, *Col1a1*-positive osteoblasts accumulated on both sides of frontal bone, and the frontal suture cells condensed between them (Figure 4E,F,I,J,M,N,Q,R). In *Cbfb2*^−/−^ mice at P7, the frontal suture was more widely open in the micro-CT image (Figure 4C,D) and the condensation of the suture cells was not apparent (Figure 4G,H,K,L,O,P,S,T). In wild-type mice at P10, the frontal suture was still not completely closed in the micro-CT image (Figure 5A,B) and some chondrocytes in the suture expressed *Col2a1*, but most of them were hypertrophic chondrocytes, which expressed *Col10a1* (Figure 5E,F,I,J,M,N,Q,R,U,V). In *Cbfb2*^−/−^ mice at P10, the frontal suture was still apparently open in the micro-CT image (Figure 5C,D), frontal suture cells condensed, most of them expressed *Col2a1* and a few expressed *Col10a1* (Figure 5G,H,K,L,O,P,S,T,W,X). Thus, there were more *Col2a1*-positive cells and less *Col10a1*-positive cells in *Cbfb2*^−/−^ mice compared with wild-type mice, indicating that the process of endochondral ossification was delayed in *Cbfb2*^−/−^ mice. In wild-type mice at P14, the frontal suture was nearly closed in the micro-CT image (Figure 6A,B) and the cartilaginous tissue in the suture was replaced with bone, which was surrounded by *Col1a1*-positive osteoblasts (Figure 6E,F,I,J,M,N,Q,R,U,V). In *Cbfb2*^−/−^ mice at P14, the frontal suture was still open in the micro-CT image (Figure 6C,D), and the chondrocytes in the suture were hypertrophic and expressed *Col10a1* (Figure 6G,H,K,L,O,P,S,T,W,X). At P28, the frontal suture was completely closed in both wild-type and *Cbfb2*^−/−^ mice in the micro-CT images (Figure 7A–D) and the suture was also completely closed in the histological analysis in both groups (Figure 7E–H). Thus, the closure process of the posterior frontal suture was delayed, but not interrupted in *Cbfb2*^−/−^ mice. Incisor development was similar between wild-type and *Cbfb2*^−/−^ mice according to the micro-CT analyses (Figure 4A,C, Figure 5A,C, Figure 6A,C and Figure 7A,C). The same findings were confirmed in all mice analyzed. 

### 2.4. Shortened Limb Bones, Rib Deformity and Abnormal Spinal Curvature in Cbfb2^−/−^ Mice

At 4 weeks of age, the body weight was reduced in *Cbfb2*^−/−^ and *Runx2*^+/−^ mice compared with the respective control groups (Figure 8A,B). The lengths of the limb bones, including the ulna, femur and tibia, were significantly reduced in *Cbfb2*^−/−^ mice, but the reduction was not significant in *Runx2*^+/−^ mice compared with the respective control mice (Figure 8C–J). No apparent differences were observed in *Cbfb2*^−/−^ and *Runx2*^+/−^ mice compared with the respective control mice in histological analysis of femurs (Appendix A). Rib deformities were observed in all *Cbfb2*^−/−^ mice (8/8 mice), half of which were severe, and the area of the thoracic cage in *Cbfb2*^−/−^ mice was reduced compared with that in wild-type mice (Figure 8K,L,O,P,S). In *Cbfb2*^−/−^ mice with severe rib deformity, the expression of *Col2a1* and *Col10a1*, which are expressed in immature and mature chondrocytes, respectively [30], but not *Mmp13*, which is highly expressed in terminal hypertrophic chondrocytes [30], in the ribs was extremely higher than that in wild-type mice and *Cbfb2*^−/−^ mice with mild rib deformity, except for the part of permanent cartilage (Figure 8T). However, the expression of osteoblast marker genes, including *Runx2*, *Sp7*, *Alpl*, *Spp1*, *Ibsp* and *Col1a1*, was not significantly affected (Figure 8T), indicating that endochondral ossification was impaired in *Cbfb2*^−/−^ mice with severe rib deformity due to the inhibited chondrocyte maturation. Half of the *Cbfb2*^−/−^ mice showed abnormal spinal curvature (4/8 mice), and the lengths of the 1st lumbar vertebrae in *Cbfb2*^−/−^ mice were shorter than those in wild-type mice (Figure 8O,P,U). On the other hand, the expression of these genes, except *Runx2* in *Runx2*^+/−^ mice, was not significantly different from that in wild-type mice (Appendix A). Furthermore, there was no rib deformity or abnormal spinal curvature in *Runx2*^+/−^ mice (0/7 mice), although the lengths of the 1st lumbar vertebrae were shorter than those in wild-type mice (Figure 8M,N,Q,R,S,U).

At 10 weeks of age, the body weight in *Cbfb2*^−/−^ mice was similar to the control mice, but that in *Runx2*^+/−^ mice was reduced compared with the control mice (Figure 9A,B). Suture and fontanelles were opened in *Runx2*^+/−^ mice but not in *Cbfb2*^−/−^ mice (Figure 9C–G), and the length of clavicles was mildly and severely reduced in *Cbfb2*^−/−^ and *Runx2*^+/−^ mice, respectively, at 14 weeks of age (Figure 9H–L). Although the rib deformity was observed in half of the *Cbfb2*^−/−^ mice (7/14 mice), it was mild and the area of the thoracic cage was similar to that in wild-type mice (Figure 9H,I,M). There was no rib deformity in *Runx2*^+/−^ mice, and the area of thoracic cage was similar to that in wild-type mice (Figure 9J,K,M). The abnormal spinal curvature was observed in about one-third of the *Cbfb2*^−/−^ mice (5/14 mice). The frequencies of rib deformity and abnormal spinal curvature were two-thirds (22/33 mice) and one-third (12/33 mice), respectively, in *Cbfb2*^−/−^ mice between 4 weeks and 10 months of age, whereas these deformities were not observed in *Runx2*^+/−^ mice (0/26 mice) and wild-type mice (0/43 mice). Furthermore, severe rib deformities were not observed in the *Cbfb2*^−/−^ mice after 10 weeks of age, suggesting that severe rib deformity is involved, at least in part, in being a cause of death in *Cbfb2*^−/−^ mice after birth.

### 2.5. Micro-CT Analysis of Femurs and Vertebrae in Cbfb2^−/−^ and Runx2^+/−^ Mice at 4 Weeks and 10 Weeks of Age

From micro-CT analysis, the trabecular bone volume, trabecular thickness and trabecular number in *Cbfb2*^−/−^ femurs were similar, but the trabecular bone mineral density was reduced compared with those in wild-type femurs at 4 weeks of age (Figure 10A,B). In contrast, all of the trabecular parameters were reduced in *Runx2*^+/−^ femurs compared with those in wild-type femurs (Figure 10C,D). In the analysis of femoral cortical bone, the cortical area, cortical thickness and bone mineral density were reduced in both *Cbfb2*^−/−^ and *Runx2*^+/−^ mice compared with those in the respective control mice, but with more reduction in *Cbfb2*^−/−^ mice (Figure 10E–H). The periosteal perimeter was increased in *Cbfb2*^−/−^ mice but not in *Runx2*^+/−^ mice, and the endosteal perimeter was increased in *Cbfb2*^−/−^ and *Runx2*^+/−^ mice, with a greater increase in *Cbfb2*^−/−^ mice compared with those in the respective control mice (Figure 10E–H). 

Bone volume, trabecular thickness and trabecular number in the 1st lumbar vertebrae were increased in *Cbfb2*^−/−^ mice compared with those in wild-type mice, but their values in *Runx2*^+/−^ mice were similar to those in wild-type mice at 4 weeks of age (Figure 10I–L). The trabecular bone mineral density was significantly reduced in the *Runx2*^+/−^ mice, but not in the *Cbfb2*^−/−^ mice (Figure 10J,L). 

The trabecular bone volume, trabecular thickness, trabecular number and trabecular bone mineral density were comparable in *Cbfb2*^−/−^ femurs but reduced in *Runx2*^+/−^ femurs compared with those in the respective control mice at 10 weeks of age (Figure 11A–D). Cortical area, cortical thickness and cortical bone mineral density were reduced and the endosteal perimeter was increased in the *Cbfb2*^−/−^ and *Runx2*^+/−^ femurs compared with those in the respective control mice (Figure 11E–H). The periosteal perimeter was increased in the *Cbfb2*^−/−^ femurs, but not in the *Runx2*^+/−^ femurs (Figure 11E–H). 

In the 6th lumbar vertebrae of *Cbfb2*^+/+^ and *Cbfb2*^−/−^ mice and 1th lumbar vertebrae of *Runx2*^+/+^ and *Runx2*^+/−^ mice, the bone volume of *Cbfb2*^−/−^ mice and *Runx2*^+/−^ mice were not significantly affected and the trabecular thickness was reduced in the *Runx2*^+/−^ mice, but not in the *Cbfb2*^−/−^ mice, and the trabecular number was increased, but the trabecular bone mineral density was reduced in the *Cbfb2*^−/−^ and *Runx2*^+/−^ mice compared with those in the respective control mice (Figure 11I–L). 

### 2.6. Bone Histomorphometric Analysis of Femurs and Vertebrae and Serum Markers for Bone Formation and Resorption in Cbfb2^−/−^ and Runx2^+/−^ Mice at 10 Weeks of Age

In the bone histomorphometric analysis of femurs, the osteoblast parameters, including osteoid surface, osteoid thickness, osteoblast surface and osteoblast number, were reduced in the *Runx2*^+/−^ mice, but not in the *Cbfb2*^−/−^ mice, compared with those in the respective control mice (Figure 12A–D). Osteoclast surface in the *Cbfb2*^−/−^ mice and osteoclast number in the *Runx2*^+/−^ mice were reduced, and the mineral apposition rate and bone formation rate were reduced in the *Cbfb2*^−/−^ and *Runx2*^+/−^ mice compared with those in the respective control mice (Figure 12B,D). The mineralizing surface was significantly reduced in the *Runx2*^+/−^ mice, but not in the *Cbfb2*^−/−^ mice (Figure 12B,D).

In the bone histomorphometric analysis of vertebrae, osteoblast parameters, including osteoid surface, osteoid thickness, osteoblast surface and osteoblast number, were increased in *Cbfb2*^−/−^ mice but reduced in *Runx2*^+/−^ mice compared with those in the respective control mice (Figure 12E–H). Osteoclast parameters, including osteoclast surface, osteoclast number and eroded surface, in the *Cbfb2*^−/−^ mice were similar to the control mice, but the eroded surface in *Runx2*^+/−^ mice was reduced compared with the respective control mice (Figure 12F,H). Mineral apposition rate, mineralizing surface and bone formation rate in *Cbfb2*^−/−^ mice were similar compared with the control mice but reduced in *Runx2*^+/−^ mice compared with those in the control mice (Figure 12F,H). 

In the dynamic bone histomorphometric analysis of femoral cortical bone, mineral apposition rate, mineralizing surface and bone formation rate in the periosteum and endosteum in *Cbfb2*^−/−^ mice were reduced compared with those in the control mice, whereas the mineralizing surface and bone formation rate, but not the mineral apposition rate in the periosteum, in *Runx2*^+/−^ mice were reduced compared with the control, and these parameters in the endosteum were unchanged compared with those in the control mice (Figure 13A–P). 

The serum markers for bone formation (total procollagen type 1 N-terminal propeptide: P1NP) and bone resorption (tartrate-resistant acid phosphatase 5b: TRAP5b) were reduced in the *Cbfb2*^−/−^ and *Runx2*^+/−^ mice compared with those in the respective control mice at 10 weeks of age (Figure 13Q). 

Since the mineral apposition rate, mineralizing surface and bone formation rate in the endosteum were reduced in the *Cbfb2*^−/−^ mice, but not in the *Runx2*^+/−^ mice, compared with those in the respective control mice, we also examined TRAP-positive cells in the endosteum. The number of TRAP-positive cells was reduced in the *Runx2*^+/−^ mice, but not in the *Cbfb2*^−/−^ mice, compared with those in the respective control mice (Figure 14). Thus, the dynamic bone histomorphometric parameters in the endosteum of *Runx2*^+/−^ mice were affected by the reduced bone resorption.

### 2.7. Expressions of the Genes Related to Osteoblast and Osteoclast Differentiation, RUNX Family Genes and Cbfb

Real-time reverse transcription-polymerase chain reaction (RT-PCR) was performed using RNA from the tibiae and vertebrae at 4 weeks of age (Figure 15A,B). In *Cbfb2*^−/−^ tibiae, the expressions of *Sp7*, *Alpl*, *Col1a1*, *Tnfsf11*, *Cbfb1*, *Cbfb2* and *Cbfb1+2* and the ratio of *Tnfsf11/Tnfrsf11b* were reduced, the expression of *Runx1* was increased and the expression of *Spp1*, *Bglap*&*Bglap2*, *Tnfrsf11b*, *Runx2* and *Runx3* was unchanged compared with those in wild-type mice (Figure 15A). In *Runx2*^+/−^ tibiae, the expression of *Spp1* and *Runx2* and the ratio of *Tnfsf11/Tnfrsf11b* were reduced, the expression of *Cbfb2* and *Cbfb1+2* was increased and the other gene expressions were unchanged compared with those in wild-type mice (Figure 15B). 

In *Cbfb2*^−/−^ vertebrae, the expression of *Alpl*, *Cbfb1*, *Cbfb2*, *Cbfb1+2* and the ratio of *Tnfsf11/Tnfrsf11b* were reduced, the expression of *Runx1* was increased and the expression of *Sp7*, *Spp1*, *Col1a1*, *Bglap*&*Bglap2*, *Tnfsf11*, *Tnfrsf11b*, *Runx2* and *Runx3* was unchanged compared with those in wild-type vertebrae (Figure 15C). In *Runx2*^+/−^ vertebrae, the expression of *Runx2* and the ratio of *Tnfsf11/Tnfrsf11b* were reduced, but the expression of the other genes was unchanged compared with those in wild-type vertebrae (Figure 15D). 

### 2.8. The Proliferation and Apoptosis of Osteoblast-Like and Osteoprogenitor-Like Cells in Femurs and Vertebrae

To investigate why the osteoblast parameters were reduced in *Runx2*^+/−^ mice but not in *Cbfb2*^−/−^ mice in bone histomorphometric analysis of femurs (Figure 12A–D), the proliferation of osteoblast-like and osteoprogenitor-like cells was examined in the femurs. The frequencies of BrdU-positive osteoblast-like and osteoprogenitor-like cells in the trabecular and cortical bone of femurs in *Cbfb2*^−/−^ mice were comparable with those in wild-type mice, whereas those in the *Runx2*^+/−^ mice were reduced compared with those in wild-type mice at P7 (Figure 16). To investigate why the osteoblast parameters were increased in *Cbfb2*^−/−^ mice but reduced in *Runx2*^+/−^ mice in bone histomorphometric analysis of vertebrae (Figure 12E–H), the proliferation and apoptosis in osteoblast-like and osteoprogenitor-like cells were examined in the vertebrae. The frequencies of BrdU-positive osteoblast-like and osteoprogenitor-like cells were increased in *Cbfb2*^−/−^ mice but reduced in *Runx2*^+/−^ mice compared with the respective control mice at 4 weeks of age (Figure 17). The frequencies of TUNEL-positive osteoblast-like and osteoprogenitor-like cells in the *Cbfb2*^−/−^ and *Runx2*^+/−^ mice were comparable to those in the respective control mice (Appendix A).

## 3. Discussion

The *Cbfb2*^−/−^ mice expressed *Cbfb* about one-fifth of that in wild-type mice. Half of *Cbfb2*^−/−^ mice died by 4 weeks of age, but half of them with mild rib deformity survived and enabled us to examine the functions of CBFB in bone development in young and adult mice. The *Runx2*^+/−^ mice were examined in detail and compared with *Cbfb2*^−/−^ mice to evaluate the required amount of CBFB and RUNX2 in the development of calvaria, trabecular and cortical bone in femurs, vertebrae and ribs. The *Cbfb2*^−/−^ mice showed differences in the phenotypes and their severities compared with *Runx2*^+/−^ mice, and the bone volumes in each line were regulated differently among the bone tissues, indicating the different requirements of CBFB and RUNX2 for bone development and maintenance among the skeletal tissues. Although CBFB stabilizes RUNX2 protein and enhances the capacity of transcriptional activation [31], the dependency of RUNX2 on CBFB was also different among the skeletal tissues.

The development of calvaria was impaired in the *Runx2*^+/−^ mice, in which the level of RUNX2 protein was half of that in wild-type calvaria and the level of CBFB protein was like that in wild-type calvaria, while it was mildly delayed in the *Cbfb2*^−/−^ mice, in which the level of RUNX2 protein was 70%, and that of CBFB protein was one-fifth of those in wild-type calvaria (Figure 1, Figure 4, Figure 5, Figure 6, Figure 7 and Figure 9). Furthermore, the RUNX2 protein of the calvaria in the mice, in which the neo gene was inserted in the intron of RUNX2, was reduced to two-thirds of that in wild-type mice, and the closure of the posterior frontal suture was later than *Cbfb2*^−/−^ mice (Figure 4, Figure 5, Figure 6 and Figure 7) [32]. Thus, the suture closure is likely to be dependent on the level of RUNX2 protein and RUNX2 protein is relatively stable in the low amount of CBFB in calvaria, probably due to the presence of the transcription factors and/or other cofactors, which stabilize RUNX2 protein and enhance the DNA binding capacity by interacting with RUNX2. 

CBFB, its isoform CBFB2 and RUNX2 are required for chondrocyte maturation, which is an essential step for endochondral ossification, while RUNX1 induces chondrocyte proliferation and maturation and RUNX3 is involved in chondrocyte maturation [5,6,21,22,24,25,26,33,34]. The shortening of limb long bones in *Cbfb2*^−/−^ mice were more severe than those in *Runx2*^+/−^ mice, the area of the thoracic cage in *Cbfb2*^−/−^ mice but not in *Runx2*^+/−^ mice at 4 weeks of age were significantly smaller than those in wild-type mice, and rib deformity was observed in *Cbfb2*^−/−^ mice but not in *Runx2*^+/−^ mice (Figure 8). Furthermore, the expression of chondrocyte marker genes, including Col2a1 and Col10a1, in the mineralized parts of ribs in *Cbfb2*^−/−^ mice but not in *Runx2*^+/−^ mice was markedly higher than that in wild-type mice, and the expression levels were dependent on the severity of rib deformity (Figure 8T), indicating that the inside of the mineralized bone collar of ribs in *Cbfb2*^−/−^ mice with severe rib deformity was still largely cartilaginous. These findings suggest that the process of endochondral ossification was delayed more severely in *Cbfb2*^−/−^ mice than in *Runx2*^+/−^ mice. Furthermore, the protein levels of RUNX1, RUNX2 and RUNX3 in ribs of *Cbfb2*^−/−^ mice were 11%, 25% and 19% of those in wild-type mice, respectively, although RUNX1 reduction may have been partly intensified by a lower amount of hematopoietic cells that highly express RUNX1, as endochondral ossification was retarded in the ribs but not in the other skeletons of *Cbfb2*^−/−^ mice [26] (Figure 1F,G and Figure 8T, Appendix A). Thus, the rib deformity in *Cbfb2*^−/−^ mice is likely to be caused by the impaired endochondral ossification due to the severe reduction in all RUNX family proteins. Moreover, the expression levels of CBFB and RUNX2 were lowest in rib among calvaria, tibia, vertebra and ribs (Appendix A). It may also contribute to the impaired rib development in *Cbfb2*^−/−^ mice. In contrast, vertebrae were shortened in both *Cbfb2*^−/−^ and *Runx2*^+/−^ mice (Figure 8U). RUNX2 plays a major role in chondrocyte proliferation and maturation, and CBFB also induces chondrocyte proliferation and maturation by stabilizing RUNX proteins and enhancing their DNA binding [31]. Therefore, RUNX2 and CBFB cooperatively accelerate the anterior-posterior axis development by regulating chondrocyte proliferation and maturation.

The trabecular bone in femurs was reduced in *Runx2*^+/−^ mice but not in *Cbfb2*^−/−^ mice in micro-CT analysis (Figure 10A–D and Figure 11A–D), osteoblast parameters were reduced in *Runx2*^+/−^ mice but not in *Cbfb2*^−/−^ mice in bone histomorphometric analysis, and the bone formation rate was reduced in both *Cbfb2*^−/−^ and *Runx2*^+/−^ mice, but more severely in *Runx2*^+/−^ mice (Figure 12A–D). Furthermore, the serum marker for bone formation, P1NP, was reduced more apparently in *Runx2*^+/−^ mice than in *Cbfb2*^−/−^ mice. (Figure 13Q). Therefore, the reduction in trabecular bone in *Runx2*^+/−^ mice but not in *Cbfb2*^−/−^ mice is likely to be explained by the level of the reduction in bone formation, as bone resorption was similarly reduced in *Cbfb2*^−/−^ and *Runx2*^+/−^ mice (Figure 12B,D and Figure 13Q). However, the mechanism of the reduction in bone formation was different between *Cbfb2*^−/−^ and *Runx2*^+/−^ mice. Osteoblast marker gene expression was reduced in *Cbfb2*^−/−^ mice but not in *Runx2*^+/−^ mice in real-time RT-PCR analysis (Figure 15A,B), while the number of osteoblasts and BrdU-positive cells in osteoblast-like cells were reduced in the trabecular bone of femurs in *Runx2*^+/−^ but not in *Cbfb2*^−/−^ mice (Figure 12B,D and Figure 16). These findings suggest that osteoblast maturation was impaired, but the number of osteoblasts was maintained in *Cbfb2*^−/−^ mice, while the number of osteoblasts was reduced but osteoblast maturation was unaffected in *Runx2*^+/−^ mice. This is consistent with the previous report, which showed that the commitment of mesenchymal cells to osteoblasts and the proliferation of osteoblast progenitors are impaired in *Runx2*^+/−^ mice, but osteoblast marker gene expression in mineralized calvaria are normal in *Runx2*^+/−^ mice [27]. As the levels of RUNX2 protein were about half (54%) and RUNX1 and RUNX3 proteins were also reduced to 61% and 25%, respectively, in the trabecular bone of femurs and tibiae in *Cbfb2*^−/−^ mice (Figure 1D,E), other cofactors or transcription factors may have compensated for the deficiency of *Cbfb* and have strengthened the activity of RUNX family proteins, especially RUNX2, in the trabecular bone. 

In contrast to the trabecular bone in femurs, cortical bone in femurs was reduced in both *Cbfb2*^−/−^ and *Runx2*^+/−^ mice in micro-CT analysis (Figure 10E–H and Figure 11E–H). Although bone formation rate in the periosteum was reduced in both *Cbfb2*^−/−^ and *Runx2*^+/−^ mice, that of the endosteum was reduced in *Cbfb2*^−/−^ mice but not in *Runx2*^+/−^ mice in bone histomorphometric analysis (Figure 13M–P). Although osteoclast parameters were reduced in the trabecular bone in both *Cbfb2*^−/−^ and *Runx2*^+/−^ mice (Figure 12B,D), the number of TRAP-positive cells in the endosteum was reduced in *Runx2*^+/−^ mice but not in *Cbfb2*^−/−^ mice compared with that in the respective control mice (Figure 14). Furthermore, the endosteal perimeter was more markedly enlarged in *Cbfb2*^−/−^ mice than *Runx2*^+/−^ mice (Figure 10E–H and Figure 11E–H). Thus, the bone resorption in the endosteum was more in *Cbfb2*^−/−^ mice than in *Runx2*^+/−^ mice, and it likely affected the bone formation rate in the endosteum. Furthermore, as the eroded surface was reduced in vertebrae in *Runx2*^+/−^ mice but not in *Cbfb2*^−/−^ mice (Figure 12F,H), the levels of the reduction of bone resorption were different not only between the trabecular and cortical bone, but also among the bone tissues in *Cbfb2*^−/−^ and *Runx2*^+/−^ mice, although the serum TRAP5b was similarly reduced in *Cbfb2*^−/−^ and *Runx2*^+/−^ mice (Figure 13Q). 

The volume of the trabecular bone in femurs and vertebrae was differentially regulated in both *Cbfb2*^−/−^ and *Runx2*^+/−^ mice. The trabecular bone volume in femurs was normal but increased significantly at 4 weeks of age and marginally at 10 weeks of age in vertebrae in *Cbfb2*^−/−^ mice in micro-CT analysis (Figure 10B,J and Figure 11B,J). Furthermore, osteoblast parameters were increased in the vertebrae but normal in the femurs in *Cbfb2*^−/−^ mice in bone morphometric analysis (Figure 12B,F). *Alpl* expression was reduced and BrdU-positive cells were increased in the vertebrae in *Cbfb2*^−/−^ mice (Figure 15C and Figure 17I), suggesting an increase in immature osteoblasts. As osteoclast parameters were not reduced, the increase in immature osteoblasts is likely to be a cause of the increase of trabecular bone in vertebrae in *Cbfb2*^−/−^ mice. As the protein levels of RUNX1, RUNX2 and RUNX3 in *Cbfb2*^−/−^ mice were about 81%, 100% and 42% of those in wild-type mice, respectively, RUNX2 protein seemed to be stabilized by other proteins than CBFB and RUNX2 is likely to have enhanced the commitment to osteoblasts and/or enhanced the proliferation of osteoblast progenitors by interacting with the unknown proteins. In *Runx2*^+/−^ mice, the volume of trabecular bone in femurs was reduced but normal in vertebrae in micro-CT analysis (Figure 10D,L and Figure 11D,L). The parameters for osteoblasts, osteoclasts and bone formation in bone morphometric analysis, the ratio of *Tnfsf11/Tnfrsf11b* expression and BrdU-positive cells were reduced in the trabecular bone in both femurs and vertebrae in *Runx2*^+/−^ mice (Figure 12D,H, Figure 15B,D, Figure 16J and Figure 17I). Therefore, the difference in the trabecular bone volume between femurs and vertebrae was likely due to the levels of reduction in bone formation and resorption.

## 4. Conclusions

In the *Cbfb2*^−/−^ and *Runx2*^+/−^ mice, the bone volumes were regulated differently among bone tissues and between the trabecular and cortical bone. The differences were likely due to the balance of bone formation and resorption, which was different among bone tissues. Calvaria and clavicles were the exceptions. More than half the dosage of RUNX2 was required for the development. The protein levels of the RUNX family and CBFB, which determine the proliferation and differentiation of chondrocytes and osteoblasts, were different among bone tissues in the *Cbfb2*^−/−^ mice. The differences in the protein levels seemed to be caused by the amount of cofactors or transcription factors in the bone tissues, which can compensate for the deficiency of CBFB, and the amount of these factors is likely to determine the level of dependency of RUNX proteins on CBFB in the bone tissues. These cofactors or transcription factors need to be identified. The current study also indicated the importance of examining multiple skeletal tissues to evaluate the functions of target genes in bone development and maintenance.

## 5. Materials and Methods

### 5.1. Mice

*Cbfb1*^−/−^, *Cbfb2*^−/−^ and *Runx2*^+/−^ mice were generated as previously described [35,36]. The backgrounds of *Cbfb1*^−/−^ and *Cbfb2*^−/−^ mice were a mixed 129Ola/C57BL6 background, and that of *Runx2*^+/−^ mice was originally a mixed 129Ola/C57BL6 background and then backcrossed with C57BL/6N at least 12 times before this study. Prior to the investigation, all the experimental protocols were reviewed and approved by the Animal Care and Use Committee of Nagasaki University Graduate School of Biomedical Sciences (No. 1403111129-21). Animals were housed three per cage in a pathogen-free environment on a 12-h light cycle at 22 ± 2 °C, with standard chow (CLEA Japan, Tokyo, Japan) and free access to tap water.

### 5.2. Skeletal and Micro-CT Analyses

Whole skeletons were stained with alcian blue and alizarin red, as described previously [36]. Micro-CT analysis was performed using a micro-CT system (R_mCT; Rigaku Corporation, Tokyo, Japan). Data from the scanned slices were used for three-dimensional analysis to calculate femoral morphometric parameters. Trabecular bone parameters were measured on a distal femoral metaphysis. Craniocaudal scans of approximately 2.4 mm (0.5 mm far from the growth plate), for 200 slices in 12-μm increments, were taken. The cortical bone parameters were measured in the mid-diaphysis of the femurs. The trabecular bone parameters in the vertebrae were measured in the 1st or 6th lumbar vertebrae. The threshold of the mineral density was 500 mg/cm^3^ in adult mice and 400 mg/cm^3^ in mice at 4 weeks of age.

### 5.3. Histological Analyses

Mice were fixed in 4% paraformaldehyde/0.1 M phosphate buffer and embedded in paraffin. Sections of 4 μm in thickness were stained with hematoxylin and eosin (H–E) or tartrate-resistant acid phosphatase (TRAP). For safranin O staining, the sections were stained with hematoxylin, fast green and safranin O. In-situ hybridization was conducted using mouse *Col2a1*, *Col10a1* and *Col1a1* antisense and sense probes, as described previously [37]. The sections were counterstained with methyl green. To analyze BrdU incorporation, we intraperitoneally injected BrdU into mice at 100 μg/g body weight 1 h before sacrifice and detected BrdU incorporation using a BrdU staining kit (Invitrogen, Carlsbad, CA, USA). The sections were counterstained with hematoxylin. TUNEL staining was performed using the ApopTag Peroxidase In-Situ Apoptosis Detection kit (Sigma Aldrich, St. Louis, MO, USA). The sections were counterstained with methyl green.

### 5.4. Bone Histomorphometric Analysis

Mice were intraperitoneally injected with calcein 7 and 2 days before sacrifice at a dose of 20 mg/kg body weight, and analyzed at 10 weeks of age. Mice were euthanized and the femurs, and lumbar vertebrae (L3–L5) were harvested and fixed in 70% ethanol for 3 days. Fixed bones were dehydrated with graded ethanol and infiltrated and embedded in the mixture of methyl methacrylate and 2-hydroxyethyl methacrylate (Fujifilm Wako pure chemical, Osaka, Japan). The bone histomorphometric analysis was performed in distal femurs and lumbar vertebrae using undecalcified 4-μm-thick sections as previously described [30]. The bone histomorphometric analysis of cortical bone was performed using 20-μm cross-sections from mid-diaphyses of femurs. The structural, dynamic and cellular parameters were calculated and expressed according to the standard nomenclature [38].

### 5.5. Serum Testing

The serum levels of total P1NP and TRAP5b were measured using the Rat/Mouse P1NP ELISA kit (Immunodiagnostic Systems, Boldon, UK) and Mouse TRAP assay (Immunodiagnostic Systems), respectively.

### 5.6. Real-Time RT-PCR and Western Blot Analyses

Total RNA was extracted using ISOGEN (Wako, Osaka, Japan). Real-time RT-PCR was performed using a THUNDERBIRD SYBR qPCR Mix (Toyobo, Osaka, Japan) and Light Cycler 480 real-time PCR system (Roche Diagnostics, Tokyo, Japan). Primer sequences are shown in Appendix A. The primer set for *Bglap*&*Bglap2* amplifies both genes and that for *Cbfb1+2* amplifies both isoforms. The expression of *Tnfsf11* and *Tnfsf11b* was examined using TaqMan probes, Mm00441906_m1 and Mm1205928_m1 (Thermo Fisher Scientific, Tokyo, Japan), respectively. The values were normalized with those of *Actb* and *Actb* using Taqman probes, Mm02619580_g1 (Thermo Fisher Scientific), respectively. Western blot was performed using mouse monoclonal anti-CBFB [26], rabbit polyclonal anti-RUNX1 [24,26], anti-RUNX2 (Cell Signaling, Danvers, MA, USA), anti-RUNX3 (Cell Signaling) and anti-β-ACTIN (Santa Cruz Biotechnology, Dallas, TX, USA) antibodies.

### 5.7. Statistical Analysis

Values are shown as the mean ± SD. Statistical analyses were performed using the Student’s *t*-test. A *p*-value less than 0.05 was considered significant. *p*-values are indicated in the graphs by * as * *p* < 0.05, ** *p* < 0.01 and *** *p* < 0.001, and those of more than three groups were performed by ANOVA and the Tukey-Kramer post-hoc test. A *p* value < 0.05 was considered significant.

## Figures and Tables

**Figure 1 ijms-23-13299-f001:**
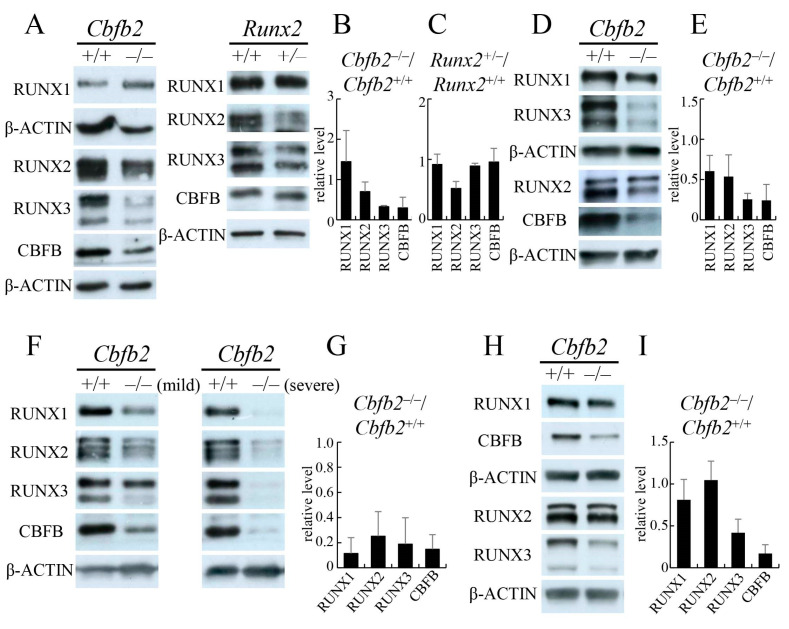
Western blot analysis of RUNX family proteins and CBFB. Proteins were extracted from calvaria (**A**–**C**), the trabecular bone in femurs and tibiae (**D**,**E**), ribs (**F**,**G**) and vertebrae (**H**,**I**) at 4 weeks of age in *Cbfb2*^+/+^, *Cbfb2*^−/−^, *Runx2*^+/+^ and *Runx2*^+/−^ mice. Representative data are shown in (**A**), (**D**), (**F**), and (**H**). β-ACTIN was used as an internal control. The intensities of bands were normalized against β-ACTIN, and the normalized values in *Cbfb2*^+/+^ or *Runx2*^+/+^ mice were set as 1. The means ± SD of the relative levels of *Cbfb2*^−/−^ mice (**B**,**E**,**G**,**I**) and *Runx2*^+/−^ mice (**C**) are shown. The number of mice analyzed: B, n = 3–5; C, n = 3; E, n = 4–5; G, n = 4; I, n = 4.

**Figure 2 ijms-23-13299-f002:**
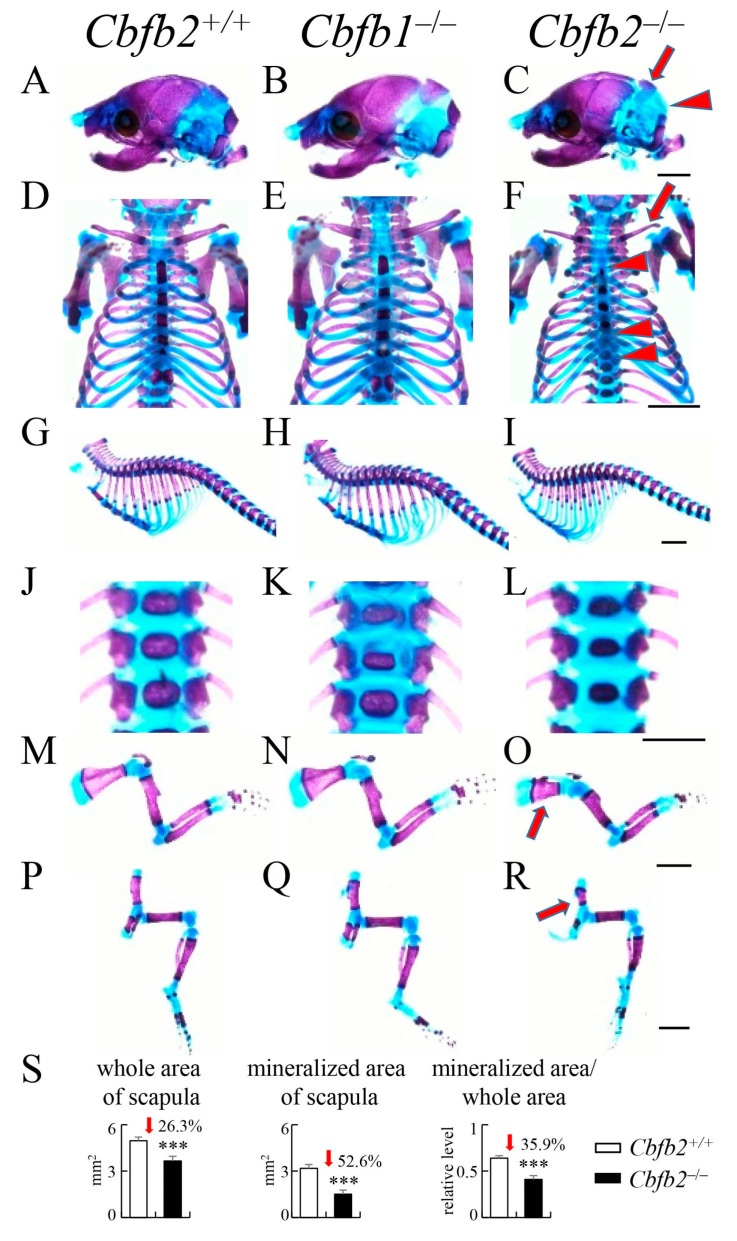
Skeletal development in *Cbfb2*^+/+^, *Cbfb1*^−/−^ and *Cbfb2*^−/−^ newborn mice. *Cbfb1*^+/+^ mice were similar to *Cbfb2*^+/+^ mice, and are not shown here. (**A**–**C**) Lateral view of skulls. (**D**–**F**) Neck and chest. (**G**–**I**) Lateral view of ribs and vertebrae. (**J**–**L**) Enlarged pictures of vertebrae. (**M**–**O**) Forelimbs. (**P**–**R**) Hind limbs. The arrow and arrowhead in (**C**) show delayed mineralization of interparietal bone and supraoccipital bone, respectively. The arrow and arrowheads in (**F**) show thin clavicle and delayed mineralization of sternum, respectively. The arrows in (**O**) and (**R**) show delayed mineralization of scapula and pelvic bones, respectively. Scale bars: 0.2 cm. (**S**) Whole area and mineralized area of scapulae and the ratios. The number of mice analyzed: *Cbfb1*^+/+^, n = 3; *Cbfb1*^−/−^, n =2; *Cbfb2*^+/+^, n = 5; *Cbfb2*^−/−^, n = 4. *** *p* < 0.001.

**Figure 3 ijms-23-13299-f003:**
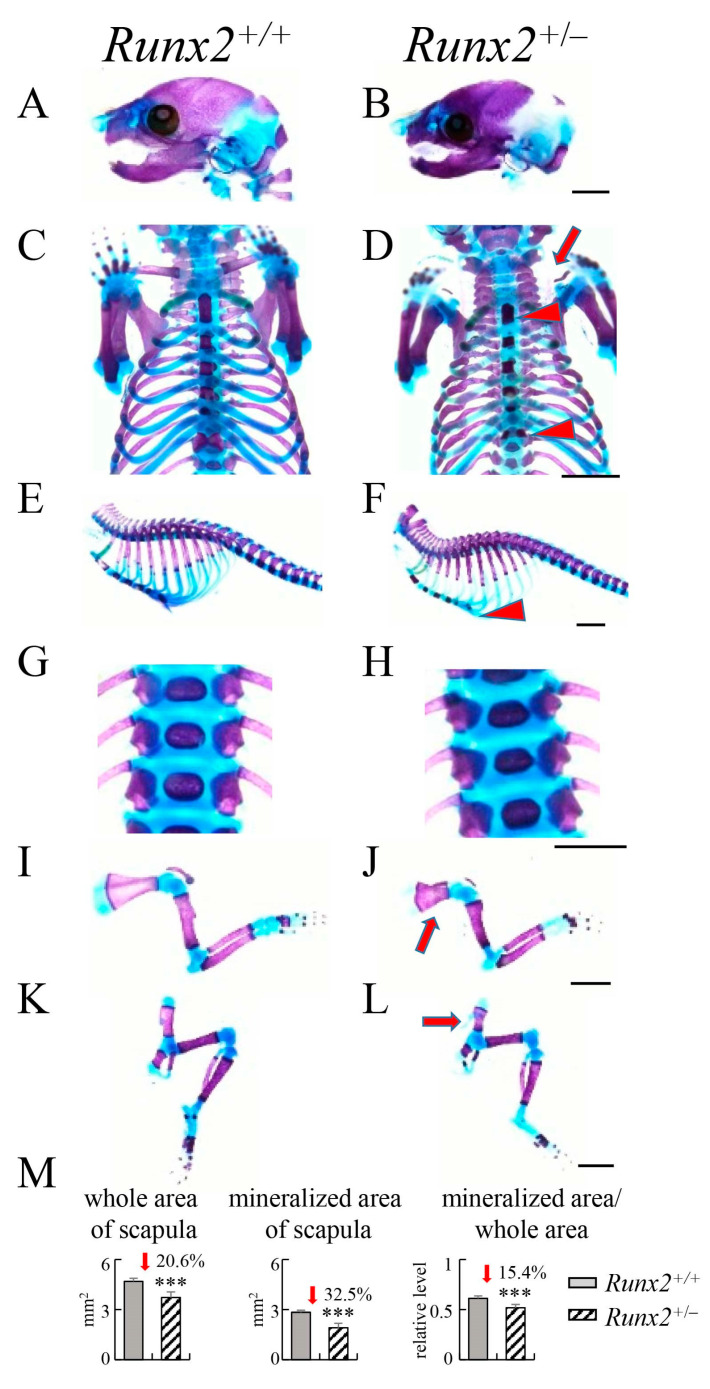
Skeletal development in *Runx2*^+/+^ and *Runx2*^+/−^ newborn mice. (**A**,**B**) Lateral view of skulls. (**C**,**D**) Neck and chest. (**E**,**F**) Lateral view of ribs and vertebrae. (**G**,**H**) Enlarged pictures of vertebrae. (**I**,**J**) Forelimbs. (**K**,**L**) Hind limbs. The arrow in (**D**) shows hypoplastic clavicle, and arrowheads in (**D**,**F**) show delayed mineralization of sternum. The arrows in (**J**,**L**) show delayed mineralization of scapula and pelvic bones, respectively. Scale bars: 0.2 cm. (**M**) Whole area and mineralized area of scapulae and the ratios. Six *Runx2*^+/+^ mice and nine *Runx2*^+/−^ mice were analyzed. *** *p* < 0.001.

**Figure 4 ijms-23-13299-f004:**
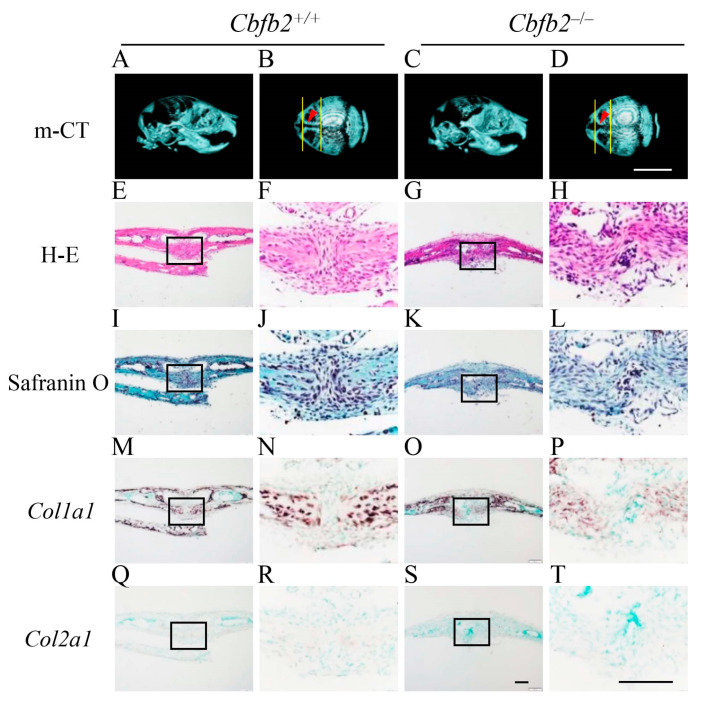
Micro-CT and histological analyses of posterior frontal sutures in *Cbfb2*^+/+^ and *Cbfb2*^−/−^ mice at P7. (**A**–**D**) Lateral (**A**,**C**) and dorsal (**B**,**D**) views of micro-CT images of skulls. The two lines in (**B**) and (**D**) show the anterior and posterior boundaries of the posterior frontal suture, and arrow heads indicate the location of coronal sections in the histological analyses. (**E**–**H**) H-E staining. (**I**–**L**) Safranin O staining. (**M**–**T**) In-situ hybridization using *Col1a1* (**M**–**P**) and *Col2a1* (**Q**–**T**) probes. The boxed regions in the left columns were magnified in the right columns. Scale bars: 0.5 cm (**A**–**D**) and 100 μm (**E**–**T**). Three *Cbfb2*^+/+^ mice and two *Cbfb2*^−/−^ mice were analyzed.

**Figure 5 ijms-23-13299-f005:**
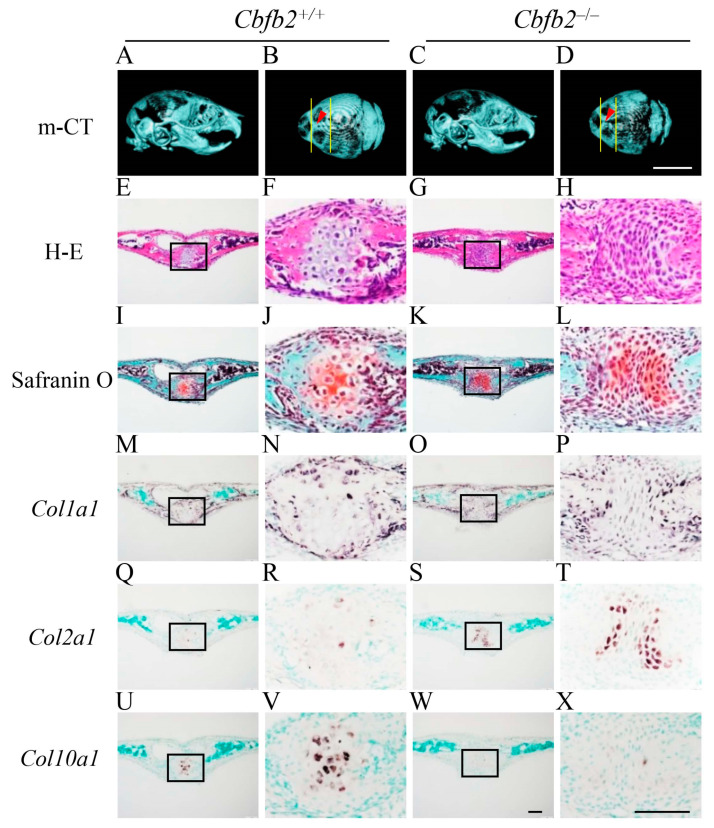
Micro-CT and histological analyses of posterior frontal sutures in *Cbfb2*^+/+^ and *Cbfb2*^−/−^ mice at P10. (**A**–**D**) Lateral (**A**,**C**) and dorsal (**B**,**D**) views of micro-CT images of skulls. The two lines in (**B**) and (**D**) show the anterior and posterior boundaries of the posterior frontal suture, and arrow heads indicate the location of coronal sections in the histological analyses. (**E**–**H**) H-E staining. (**I**–**L**) Safranin O staining. (**M**–**X**) In-situ hybridization using a *Col1a1* (**M**–**P**), *Col2a1* (**Q**–**T**) and *Col10a1* (**U**–**X**) probes. The boxed regions in the left columns were magnified in the right columns. In-situ hybridization using the sense probes showed no significant signals. Scale bars: 0.5 cm (**A**–**D**) and 100 μm (**E**–**X**). Three *Cbfb2*^+/+^ mice and two *Cbfb2*^−/−^ mice were analyzed.

**Figure 6 ijms-23-13299-f006:**
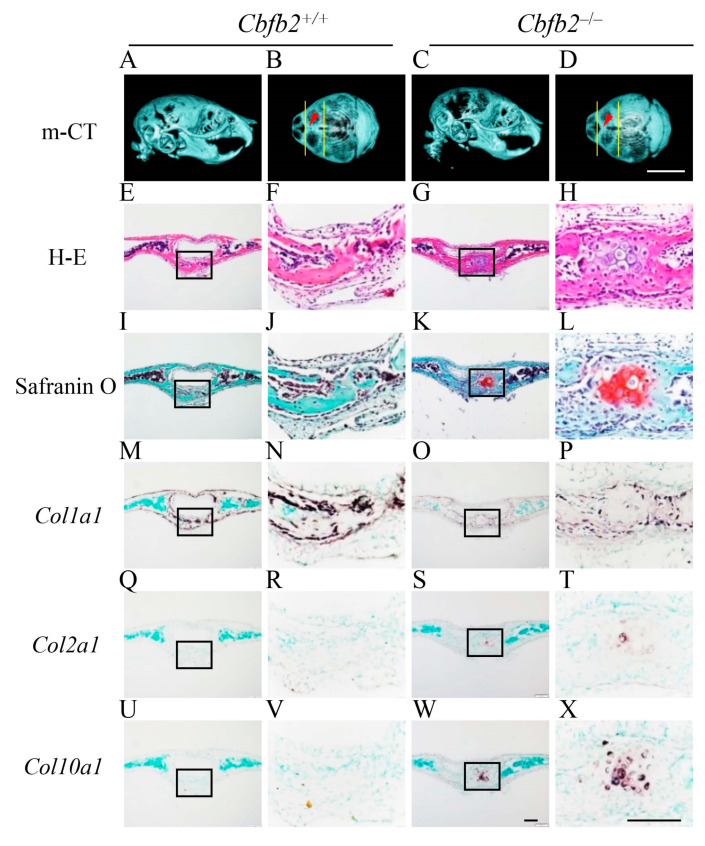
Micro-CT and histological analyses of posterior frontal sutures in *Cbfb2*^+/+^ and *Cbfb2*^−/−^ mice at P14. (**A**–**D**) Lateral (**A**,**C**) and dorsal (**B**,**D**) views of micro-CT images of skulls. The two lines in (**B**,**D**) show the anterior and posterior boundaries of the posterior frontal suture, and arrow heads indicate the location of coronal sections in the histological analyses. (**E**–**H**) H-E staining. (**I**–**L**) Safranin O staining. (**M**–**X**) In-situ hybridization using a *Col1a1* (**M**–**P**), *Col2a1* (**Q**–**T**) and *Col10a1* (**U**–**X**) probes. The boxed regions in the left columns were magnified in the right columns. Scale bars: 0.5 cm (**A**–**D**) and 100 μm (**E**–**X**). Three *Cbfb2*^+/+^ mice and two *Cbfb2*^−/−^ mice were analyzed.

**Figure 7 ijms-23-13299-f007:**
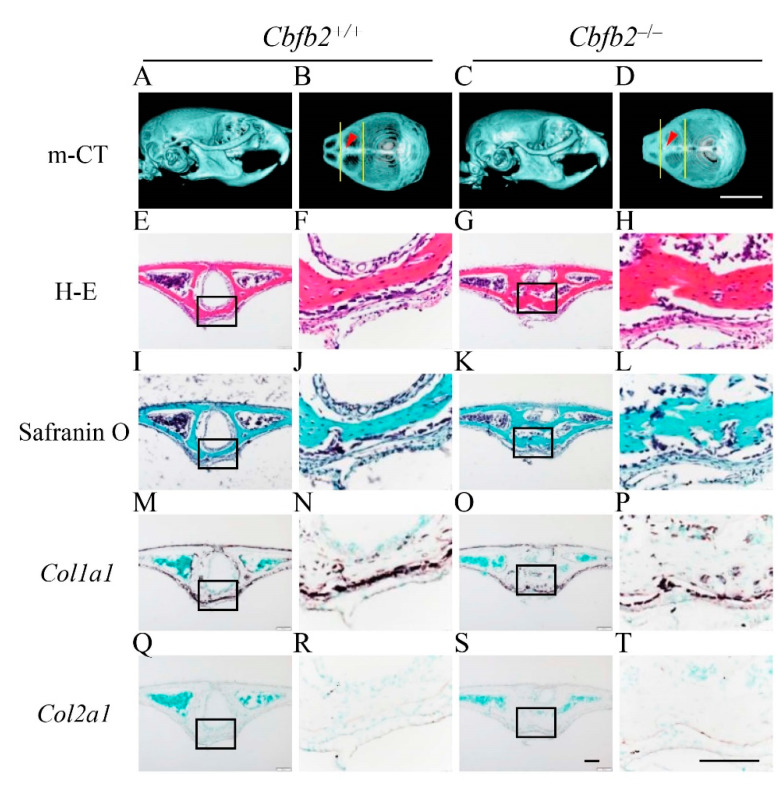
Micro-CT and histological analyses of posterior frontal sutures in *Cbfb2*^+/+^ and *Cbfb2*^−/−^ mice at P28. (**A**–**D**) Lateral (**A**,**C**) and dorsal (**B**,**D**) views of micro-CT images of skulls. The two lines in (**B**,**D**) show the anterior and posterior boundaries of the posterior frontal suture, and arrow heads indicate the location of coronal sections in the histological analyses. (**E**–**H**) H-E staining. (**I**–**L**) Safranin O staining. (**M**–**T**) In-situ hybridization using *Col1a1* (**M**–**P**) and *Col2a1* (**Q**–**T**) probes. The boxed regions in the left columns were magnified in the right columns. Scale bars: 0.5 cm (**A**–**D**) and 100 μm (**E**–**T**). Three *Cbfb2*^+/+^ and *Cbfb2*^−/−^ mice were analyzed.

**Figure 8 ijms-23-13299-f008:**
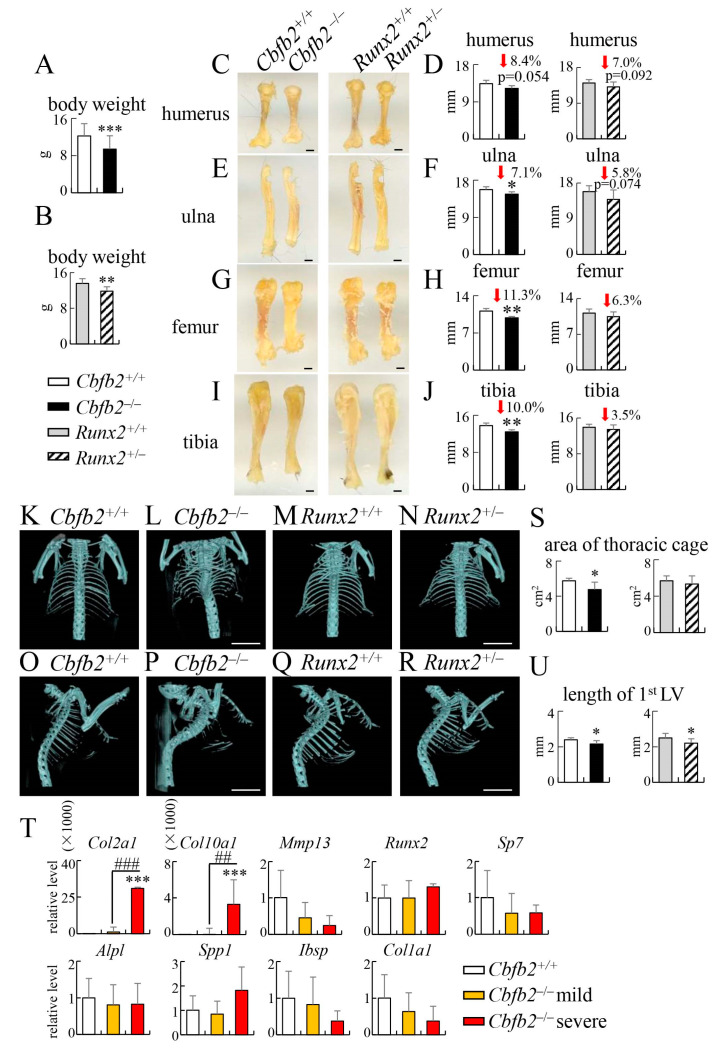
The development of limb bones, ribs and vertebrae in *Cbfb2*^−/−^ and *Runx2*^+/−^ mice at 4 weeks of age. (**A**,**B**) Body weights of male mice. n = 21–29. (**C**–**J**) The lengths of the humerus (**C**,**D**), ulna (**E**,**F**), femur (**G**,**H**) and tibia (**I**,**J**) in female mice. The number of mice analyzed: *Cbfb2*^+/+^, n = 6; *Cbfb2*^−/−^, n = 4; *Runx2*^+/+^, n = 8; *Runx2*^+/−^, n = 7. (**K**–**U**) Three-dimensional images of frontal (**K**–**N**) and lateral views (**O**–**R**) of thoracic cage and vertebrae in micro-CT, the area of thoracic cage (**S**), real-time RT-PCR analysis using bony segments of rib RNA (**T**) and the length of the 1st vertebrae (**U**) in male mice. *Cbfb2*^−/−^ mice with mild or severe rib deformity were analyzed separately in (**T**). Scale bars: 1 mm (**C**,**E**,**G**,**I**) and 1 cm (**K**–**R**). Data are shown as the mean ± SD. * *p* < 0.05, **, ## *p* < 0.01 and ***, ### *p* < 0.001. The number of mice analyzed: *Cbfb2*^+/+^, n = 4; *Cbfb2*^−/−^, n = 8 (4 mild and 4 severe); *Runx2*^+/+^, n = 8; *Runx2*^+/−^, n = 7.

**Figure 9 ijms-23-13299-f009:**
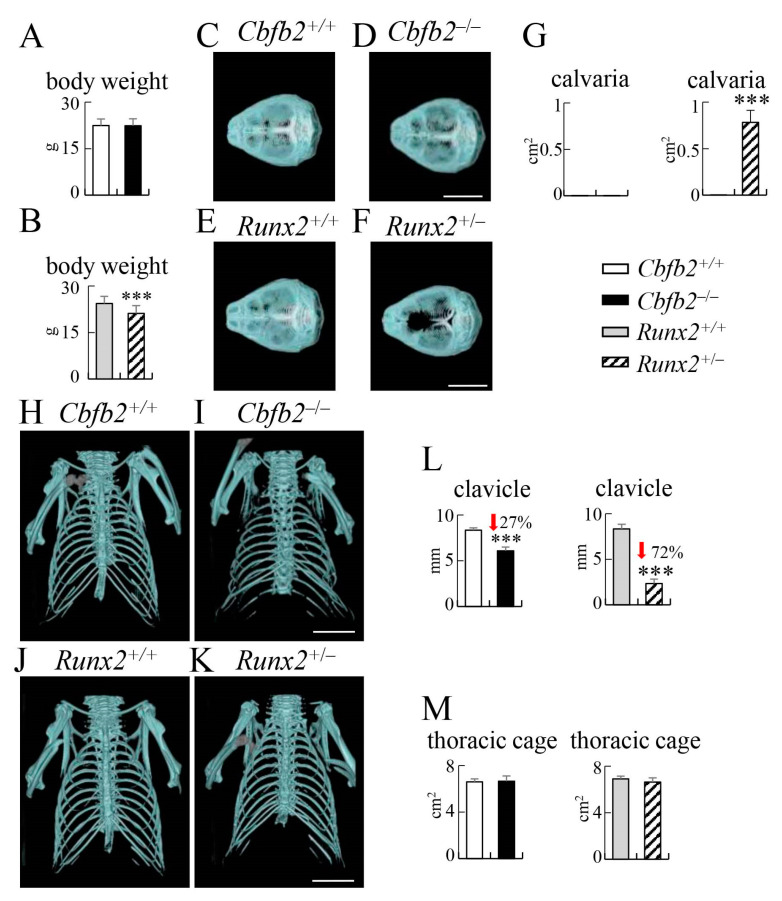
Micro-CT analysis of calvaria, clavicles and thoracic cage in *Cbfb2*^−/−^ and *Runx2*^+/−^ mice. (**A**,**B**) Body weights of male mice at 10 weeks of age. n =14–19. (**C**–**M**) Calvaria (**C**–**F**), clavicles and thoracic cage (**H**–**K**), unmineralized area in calvaria (**G**), the length of clavicles (**L**) and the area of thoracic cage (**M**) in male *Cbfb2*^+/+^, *Cbfb2*^−/−^, *Runx2*^+/+^ and *Runx2*^+/−^ mice at 14 weeks of age. Scale bars: 1 cm. Data are shown as the mean ± SD. *** *p* < 0.001. The number of mice analyzed: *Cbfb2*^+/+^, n = 7; *Cbfb2*^−/−^, n = 4; *Runx2*^+/+^, n = 5; *Runx2*^+/−^, n = 6.

**Figure 10 ijms-23-13299-f010:**
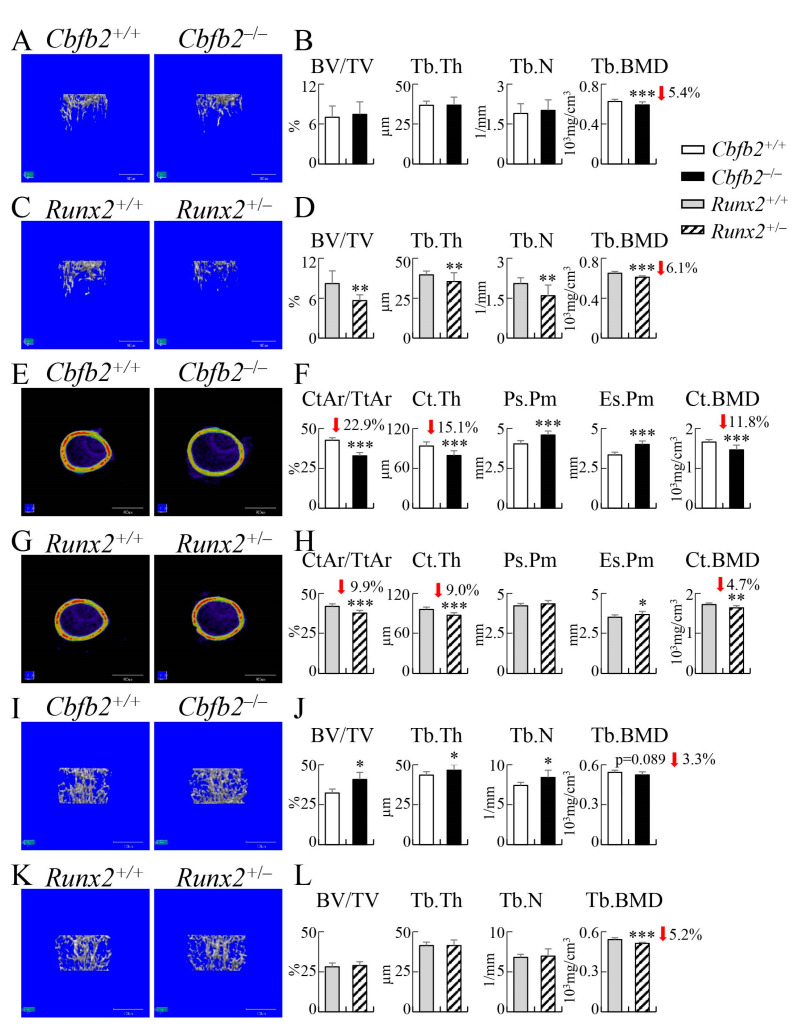
Micro-CT analyses of femurs and lumbar vertebrae in male mice at 4 weeks of age. (**A**–**D**) Three-dimensional trabecular bone architecture of distal femoral metaphysis (**A**,**C**) and quantification of the trabecular bone volume (bone volume/tissue volume, BV/TV), trabecular thickness (Tb.Th), trabecular number (Tb.N) and trabecular bone mineral density (Tb.BMD) in *Cbfb2*^+/+^ and *Cbfb2*^−/−^ mice (**B**) and *Runx2*^+/+^ and *Runx2*^+/−^ mice (**D**). (**E**–**H**) Micro-CT images of cortical bone at mid-diaphysis in femurs (**E**,**G**) and quantification of the cortical area (CtAr/TtAr), cortical thickness (Ct.Th), periosteal perimeter (Ps.Pm), endosteal perimeter (Es.Pm) and cortical bone mineral density (Ct.BMD) in *Cbfb2*^+/+^ and *Cbfb2*^−/−^ mice (**F**) and *Runx2*^+/+^ and *Runx2*^+/−^ mice (**H**). The number of mice analyzed: *Cbfb2*^+/+^, n = 22; *Cbfb2*^−/−^, n = 14; *Runx2*^+/+^, n = 11; *Runx2*^+/−^, n = 8. (**I**–**L**) Three-dimensional trabecular bone architecture of 1st lumbar vertebrae (**I**,**K**) and trabecular bone parameters in *Cbfb2*^+/+^ and *Cbfb2*^−/−^ mice (**J**) and *Runx2*^+/+^ and *Runx2*^+/−^ mice (**L**). Scale bars: 1 mm. Data are shown as the mean ± SD. * *p* < 0.05, ** *p* < 0.01 and *** *p* < 0.001. The number of mice analyzed: *Cbfb2*^+/+^, n = 9; *Cbfb2*^−/−^, n = 7; *Runx2*^+/+^, n = 11; *Runx2*^+/−^, n = 8.

**Figure 11 ijms-23-13299-f011:**
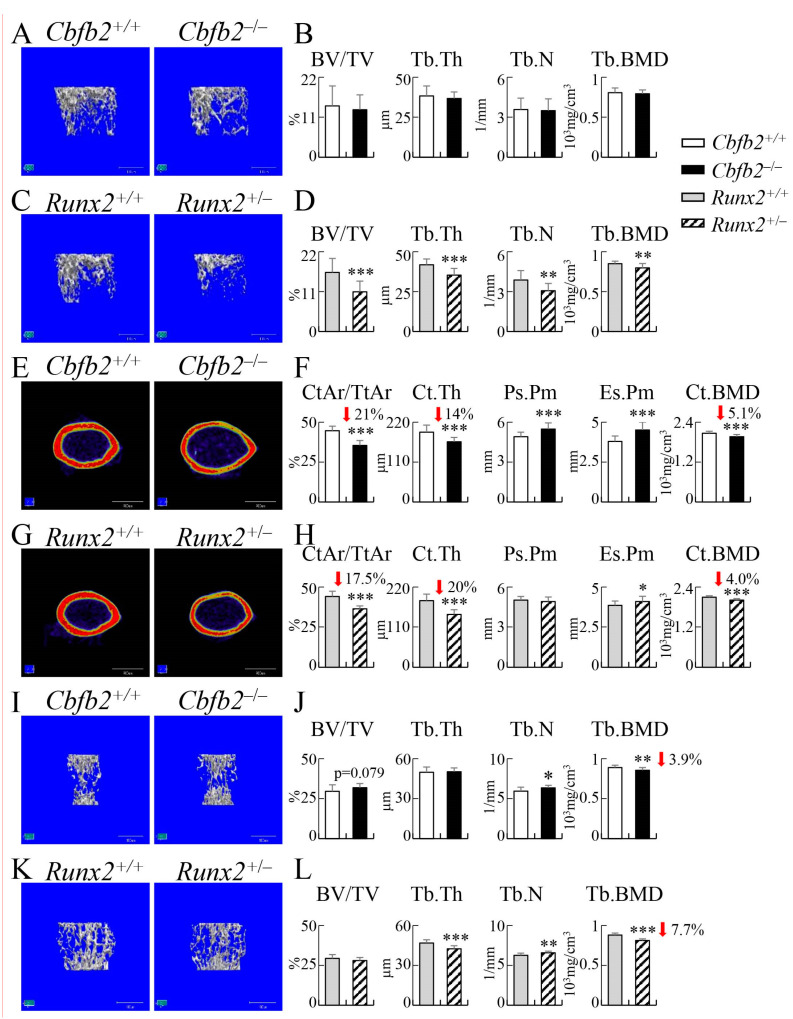
Micro-CT analyses of femurs and lumbar vertebrae in male mice at 10 weeks of age. (**A**–**D**) Three-dimensional trabecular bone architecture of distal femoral metaphysis (**A**,**C**) and quantification of the trabecular bone volume (bone volume/tissue volume, BV/TV), trabecular thickness (Tb.Th), trabecular number (Tb.N) and trabecular bone mineral density (Tb.BMD) in *Cbfb2*^+/+^ and *Cbfb2*^−/−^ mice (**B**) and *Runx2*^+/+^ and *Runx2*^+/−^ mice (**D**). (**E**–**H**) Micro-CT images of the cortical bone at mid-diaphysis in femurs (**E**,**G**) and quantification of the cortical area (CtAr/TtAr), cortical thickness (Ct.Th), periosteal perimeter (Ps.Pm), endosteal perimeter (Es.Pm) and cortical bone mineral density (Ct.BMD) in *Cbfb2*^+/+^ and *Cbfb2*^−/−^ mice (**F**) and *Runx2*^+/+^ and *Runx2*^+/−^ mice (**H**). The number of mice analyzed: *Cbfb2*^+/+^, n = 19; *Cbfb2*^−/−^, n = 14; *Runx2*^+/+^, n = 17; *Runx2*^+/−^, n = 15. (**I**–**L**) Three-dimensional trabecular bone architecture of 6th lumbar vertebrae (**I**,**K**) and trabecular bone parameters in *Cbfb2*^+/+^ and *Cbfb2*^−/−^ mice (**J**) and 1st lumbar vertebrae of those parameters in *Runx2*^+/+^ and *Runx2*^+/−^ mice (**L**). Scale bars: 1 mm (**A**,**C**,**E**,**G**) and 0.5 mm (**I**,**K**). Data are shown as the mean ± SD. * *p* < 0.05, ** *p* < 0.01 and *** *p* < 0.001. The number of mice analyzed: *Cbfb2*^+/+^, n = 9; *Cbfb2*^−/−^, n = 14; *Runx2*^+/+^, n = 14; *Runx2*^+/−^, n =13.

**Figure 12 ijms-23-13299-f012:**
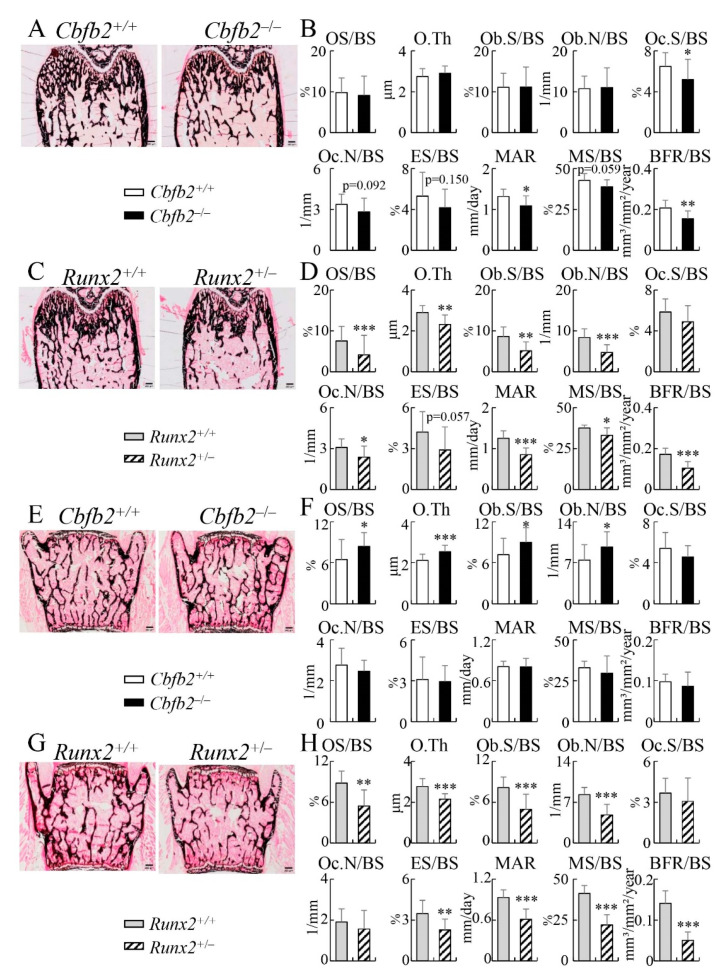
Bone histomorphometric analyses of trabecular bone in femurs and vertebrae in male mice at 10 weeks of age. (**A**–**D**) Bone histomorphometric analysis of femurs in *Cbfb2*^+/+^ and *Cbfb2*^−/−^ mice (**A**,**B**) and *Runx2*^+/+^ and *Runx2*^+/−^ mice (**C**,**D**). The osteoid surface (OS/BS), osteoid thickness (O.Th), osteoblast surface (Ob.S/BS), number of osteoblasts (N.Ob/BS), osteoclast surface (Oc.S/BS), number of osteoclasts (N.Oc/BS), eroded surface (ES/BS), mineral apposition rate (MAR), mineralizing surface (MS/BS) and bone formation rate (BFR/BS) are shown. BS, bone surface. The number of mice analyzed: *Cbfb2*^+/+^, n = 18; *Cbfb2*^−/−^, n = 14; *Runx2*^+/+^, n = 10; *Runx2*^+/−^, n = 14. (**E**–**H**) Bone histomorphometric analyses of lumbar vertebrae in *Cbfb2*^+/+^ and *Cbfb2*^−/−^ mice (**E**,**F**) and *Runx2*^+/+^ and *Runx2*^+/−^ mice (**G**,**H**). Scale bars: 0.2 mm. Data are shown as the mean ± SD. * *p* < 0.05, ** *p* < 0.01 and *** *p* < 0.001. The number of mice analyzed: *Cbfb2*^+/+^, n = 19; *Cbfb2*^−/−^, n = 14; *Runx2*^+/+^, n = 10; *Runx2*^+/−^, n = 11.

**Figure 13 ijms-23-13299-f013:**
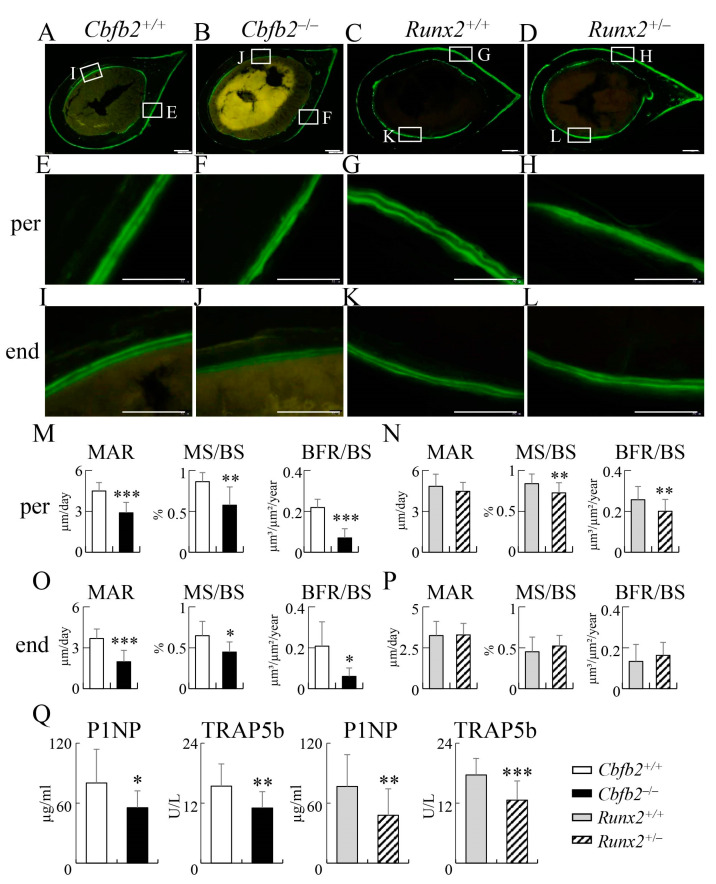
Dynamic bone histomorphometric analysis of cortical bone at the mid-diaphyses of femurs and serum markers for bone formation and resorption in male mice at 10 weeks of age. (**A**–**L**) Cross-sections of *Cbfb2*^+/+^ (**A**,**E**,**I**), *Cbfb2*^−/−^ (**B**,**F**,**J**), *Runx2*^+/+^ (**C**,**G**,**K**) and *Runx2*^+/−^ (**D**,**H**,**L**) mice, into which calcein was injected twice. The boxed regions in (**A**–**D**) are magnified in (**E**,**I**,**F**,**J**,**G**,**K**,**H**,**L**), respectively. Scale bars: 200 µm (**A**–**D**) and 100 µm (**E**–**L**). (**M**–**P**) Mineral apposition rate (MAR), mineralizing surface (MS/BS) and bone formation rate (BFR/BS) in the periosteum (**M**,**N**) and endosteum (**O**,**P**) of *Cbfb2*^+/+^ and *Cbfb2*^−/−^ mice (**M**,**O**) and *Runx2*^+/+^ and *Runx2*^+/−^ mice (**N**,**P**). *Cbfb2*^+/+^, n = 8; *Cbfb2*^−/−^, n = 8; *Runx2*^+/+^, n = 22; *Runx2*^+/−^, n = 21. (**Q**) Serum markers for bone formation (P1NP) and resorption (TRAP5b). Data are shown as the mean ± SD. * *p* < 0.05, ** *p* < 0.01 and *** *p* < 0.001. The number of mice analyzed: *Cbfb2*^+/+^, n = 14; *Cbfb2*^−/−^, n = 16; *Runx2*^+/+^, n = 20; *Runx2*^+/−^, n = 18.

**Figure 14 ijms-23-13299-f014:**
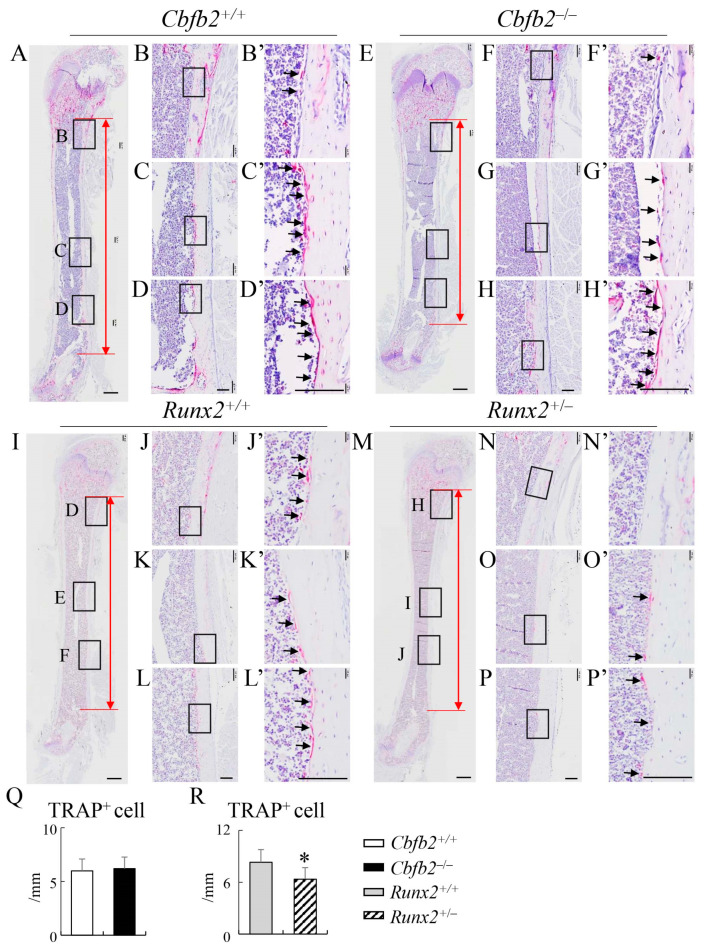
TRAP-positive cells in the endosteum of femurs at 4 weeks of age. (**A**–**N**) TRAP staining of femoral sections in *Cbfb2*^+/+^ (**A**–**D**), *Cbfb2*^−/−^ (**E**–**H**), *Runx2*^+/+^ (**I**–**L**) and *Runx2*^+/−^ (**M**–**P**) mice. The boxed regions in (**A**,**E**,**I**,**M**) are magnified in (**B**–**D**), (**F**–**H**), (**J**–**L**) and (**N**–**P**), respectively, and the boxed regions in (**B**–**D**), (**F**–**H**), (**J**–**L**) and (**N**–**P**) are magnified in (**B’**–**D’**), (**F’**–**H’**), (**J’**–**L’**) and (**N’**–**P’**), respectively. Black arrows indicate the TRAP-positive cells. Red lines show the area for counting in the endosteum. Scale bars: 500 µm (**A**,**E**,**I**,**M**) and 100 µm (**B**–**D**,**F**–**H**,**J**–**L**,**N**–**P**,**B’**–**D’**,**F’**–**H’**,**J’**–**L’**,**N’**–**P’**). (**Q**,**R**) The number of TRAP-positive cells in *Cbfb2*^+/+^ and *Cbfb2*^−/−^ mice (**Q**) and *Runx2*^+/+^ and *Runx2*^+/−^ mice (**R**). Data are shown as the mean ± SD. * *p* < 0.05. The number of mice analyzed: *Cbfb2*^+/+^ and *Cbfb2*^−/−^, n = 6; *Runx2*^+/+^ and *Runx2*^+/−^, n = 7.

**Figure 15 ijms-23-13299-f015:**
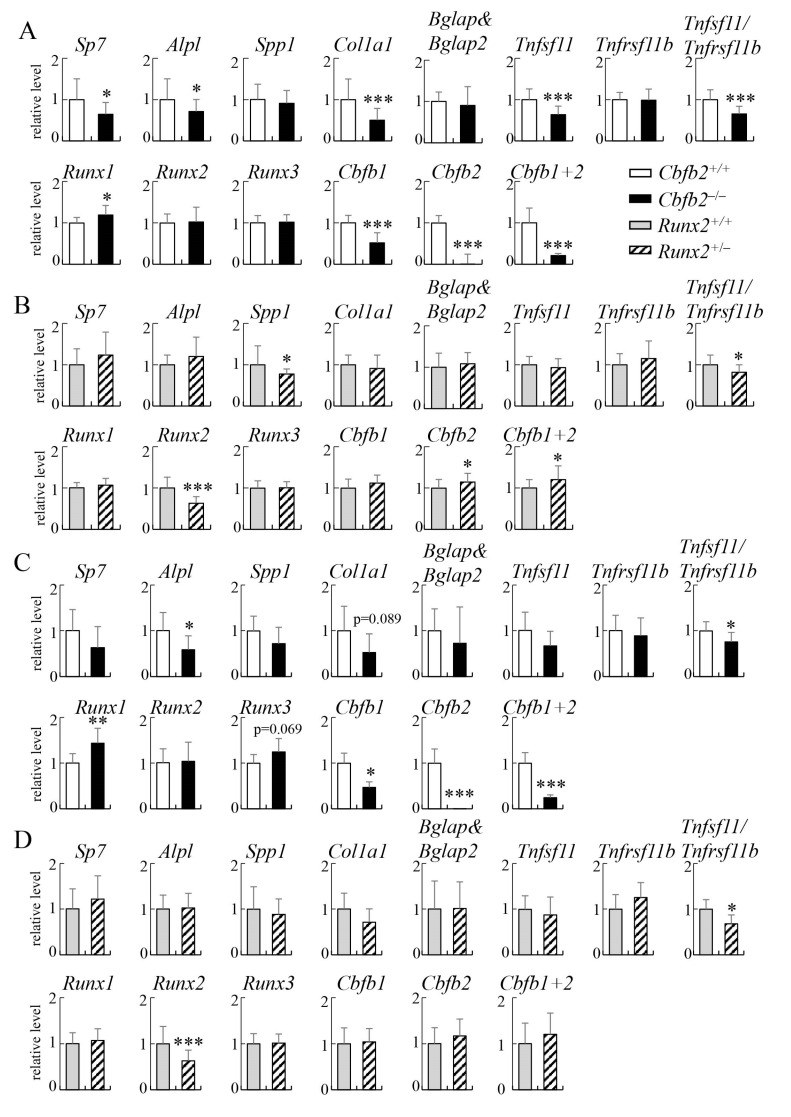
Real-time RT-PCR analyses of the expression of the genes related to osteoblast differentiation and osteoclastogenesis, RUNX family genes and *Cbfb* isoforms using RNA from tibiae and vertebrae at 4 weeks of age. Real-time RT-PCR analyses were performed using RNA from the tibiae (**A**,**B**) and vertebrae (**C**,**D**) in *Cbfb2*^+/+^ and *Cbfb2*^−/−^ mice (**A**,**C**) and *Runx2*^+/+^ and *Runx2*^+/−^ mice (**B**,**D**). The primer pair for *Bglap*&*Bglap2* amplifies both *Bglap* and *Bglap2*, and that of *Cbfb1+2* amplified both *Cbfb1* and *Cbfb2*. The ratios of *Tnfsf11* and *Tnfrsf11b* are also shown. Data are shown as the mean ± SD. * *p* < 0.05, ** *p* < 0.01 and *** *p* < 0.001. The values of *Cbfb2*^+/+^ or *Runx2*^+/+^ mice were defined as 1 and relative levels are shown. The number of mice analyzed: *Cbfb2*^+/+^, n = 16; *Cbfb2*^−/−^, n = 13; *Runx2*^+/+^, n = 19; *Runx2*^+/−^, n = 15 in the tibiae and *Cbfb2*^+/+^, n = 9; *Cbfb2*^−/−^, n = 6; *Runx2*^+/+^, n = 7; *Runx2*^+/−^, n = 7 in the vertebrae.

**Figure 16 ijms-23-13299-f016:**
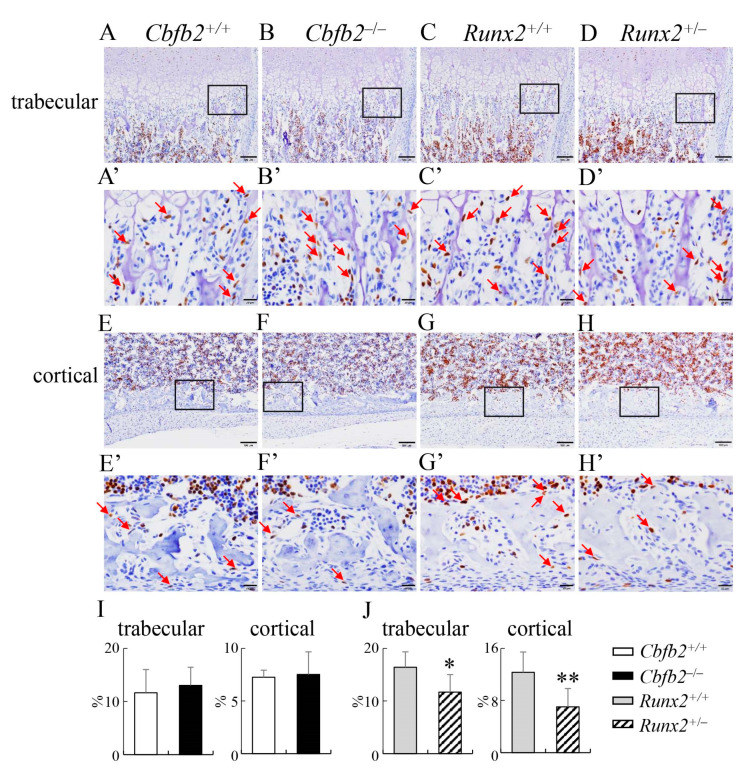
Proliferation of osteoblast-like and osteoprogenitor-like cells in the trabecular and cortical bone of femurs. (**A**–**H**) BrdU staining of trabecular bone (**A**–**D**) and cortical bone (**E**–**H**) in femurs from *Cbfb2*^+/+^ (**A**,**E**), *Cbfb2*^−/−^ (**B**,**F**), *Runx2*^+/+^ (**C**,**G**) and *Runx2*^+/−^ (**D**,**H**) mice at P7. The boxed regions in (**A**–**H**) are magnified in (**A’**–**H’**), respectively. Arrows in (**A’**–**H’**) show BrdU-positive osteoblast-like or osteoprogenitor-like cells. Scale bars: 100 µm (**A**–**H**) and 20 µm (**A’**–**H’**). (**I**,**J**) Frequencies of BrdU-positive osteoblast-like and osteoprogenitor-like cells in trabecular and cortical bone in *Cbfb2*^+/+^ and *Cbfb2*^−/−^ mice (**I**) and *Runx2*^+/+^ and *Runx2*^+/−^ mice (**J**). Data are shown as the mean ± SD. * *p* < 0.05, ** *p* < 0.01. The number of mice analyzed: *Cbfb2*^+/+^, n = 7; *Cbfb2*^−/−^, n = 5; *Runx2*^+/+^, n = 5; *Runx2*^+/−^, n = 9.

**Figure 17 ijms-23-13299-f017:**
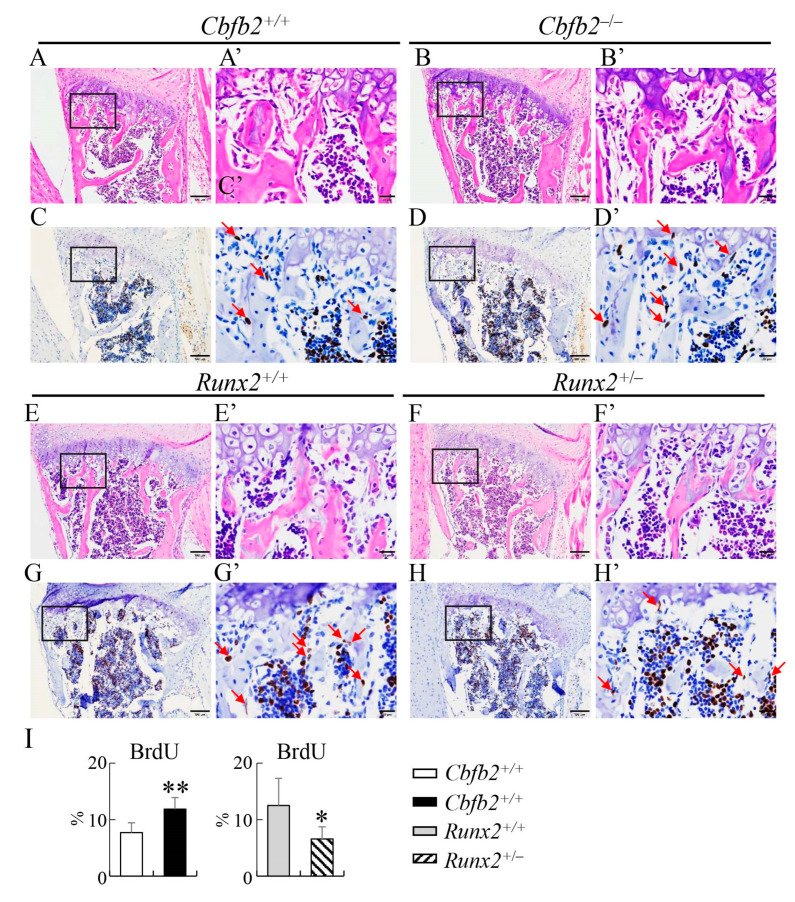
Proliferation of osteoblast-like and osteoprogenitor-like cells in vertebrae. (**A**–**H**) H-E staining (**A**,**B**,**E**,**F**) and BrdU staining (**C**,**D**,**G**,**H**) of the sections of 1st lumbar vertebrae in *Cbfb2*^+/+^ (**A**,**C**), *Cbfb2*^−/−^ (**B**,**D**), *Runx2*^+/+^ (**E**,**G**) and *Runx2*^+/−^ (**F**,**H**) mice at 4 weeks of age. The boxed regions in (**A**–**H**) are magnified in (**A’**–**H’**), respectively. Arrows in (**C’**,**D’**,**G’**,**H’**) show osteoblast-like or osteoprogenitor-like cells. Scale bars: 100 µm (**A**–**H**) and 20 µm (**A’**–**H’**). (**I**) The frequencies of BrdU-positive osteoblast-like and osteoprogenitor-like cells, which were counted in (**C’**,**D’**,**G’**,**H’**). Data are shown as the mean ± SD. * *p* < 0.05, ** *p* < 0.01. The number of mice analyzed: *Cbfb2*^+/+^, n = 3; *Cbfb2*^−/−^, n = 3; *Runx2*^+/+^, n = 3; *Runx2*^+/−^, n = 4. Two regions were counted in each mouse.

## Data Availability

Not applicable.

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
