# Peer review of "Different Requirements of CBFB and RUNX2 in Skeletal Development among Calvaria, Limbs, Vertebrae and Ribs"

_ijms, 2022, doi:10.3390/ijms232113299_

Round 1

Reviewer 1 Report

In this manuscript, the authors have analyzed the bone related phenotypic effects of a complete deletion of Cbfb2 and partial/heterozygous deletion of Runx2 in mice. They determined that both proteins regulate bone parameters differently among various bone tissues, possibly due to differential regulation of bone formation and resorption.

  1. Although the authors have extensively analyzed bone related changes upon deletion of Cbfb and partial deletion of Runx2, they have not come up with a possible unifying mechanism by which both proteins might function in regulating bone formation. Perhaps, this could be speculated on in the discussion such that the reader does not view this study as a compilation of two different studies, one relating to Cbfb and the other related to Runx2.
  2. Did the authors specifically look at alternate mechanisms of Runx2 regulation independent of Cbfb? In the case of Cbfb mediated Runx2 regulation, the authors determined that Cbfb affects Runx2 stability/expression. Runx2 being a transcription factor, is its binding to DNA affected in Cbfb knockouts? Does the existing literature provide a possible explanation in this regard?
  3. The authors demonstrate that Cbfb and Runx2 deletion affect skeletal development among calvaria, limbs, vertebrae and ribs differently. Is it possible that other transcription factors like SP7 might play a pivotal role in select skeletal bones?
  4. Does Runx2 and Cbfb expression naturally vary among the skeletal bones that the authors used in this study? If so, could this play a role in determining the basal levels of these proteins in various bone tissues, alterations of which could lead to phenotypic defects?
  5. The second paragraph of discussion reads like results section and should be modified.
  6. Conclusions: spelling error in line 573. Im-portance should be importance.

Reviewer 2 Report

In this manuscript, Jiang et al., have done an extensive analysis in Cbfb2-/- comparing them to Runx2-/- mice. Overall, the data look solid, and the manuscript is drafted well. I have concerns on the data presentation (extremely tiny figure panels), the logic and flow of the manuscript which I have outlined below. At the current format, the manuscript unfortunately warrants major revision on the presentation and the conclusion on some key aspects of the manuscript.

Major concerns:

1.       The authors present a comprehensive and extensive analysis of the phenotype in Runx2+/- and Cbfb2-/- mice. However, there must be a better logic and connection in the flow of the manuscript for easy following and readership. For instance, what is the rationale to look at osteoblast proliferation and apoptosis. Why look at Runx1, 2, and 3 at the end of the manuscript in Figure 10? Rather, the expression of Runx1, 2, 3 in Cbfa2-/- could be present as the first figure.

2.       The image panels are extremely small and printing the document on a A4 sheet, the readers will have a huge difficulty in following the data presented in the paper. Instead of putting a whole row of panels as A, I suggest that the authors split and label each panel and point out in the results what they identify from each panel. Each panel needs to be zoomed at least 4X or 5X to clearly see the data. If needed the figures can be split into multiple figures and some could be moved to supplementary. For example, for Figure 2, the authors could show stage P10 or P14 and move the rest of the individual stages to supplementary. Again, each panel needs to be labelled separately and the description should be made in the results rather than writing a result statement saying Figure 2.

3.       Figure 1C, the authors indicated that there is delayed ossification in the interparietal and supraoccipital bone in Cbfb2-/- mice with red arrows. I am not sure if that is true, as this just could be rotation or angle of the head during image acquisition. This is also can be confirmed in the Figure 1B panels of dorsal views of the Cbfb2-/- head shows normal ossification. In Figure 1, the quantification from Alcian blue/Alizarin red stained skeletal preparations may not be accurate due to the nature of the images being taken in glycerol where the samples could be in slightly orientation can have a huge reflection in the data. If the authors want to show quantification data, it must be from the µ-CT data. Authors also must state their statistical method, whether the graph is mean and SEM and what *, *** indicate?

4.       In lines 118-125, the authors point out that at P10, the wildtype mice exhibit chondrocytes in the suture that express Col2a1 but most of them express Col10a1 indicating that they are hypertrophic chondrocytes. However, these frontal bones are derived from cranial neural crest cells and undergo intramembranous bone formation rather than endochondral ossification. Therefore, the authors need to clarify their findings and in context explain how they identify endochondral ossification, ie., chondrocytes being replaced by osteoblasts, which is not true for the anterior cranial bones.

5.       The sentence regarding the weight of the mice is confusing. The weight of the mice may have nothing to do with histological analyses data. It is apparent that the length and size of the bones are affected and thus that could have implications in the weight. The authors should avoid confusing statements in lines 143-144 versus 144-145.

6.       Since the authors point out that CBFB proteins are required for the stability and DNA binding of the RUNX proteins. The authors should attempt to discuss the expectations on the phenotype of compound mutants of Cbfb2-/- Runx2+/- in various regions including the calvaria, thoracic, and long bones. In other words, how CBFA and Runx2 function together to coordinate bone development along the anterior-posterior body axis.

Minor concerns:

1.       Line 35 the list of bone matrix protein genes regulated by Runx2 is not complete. The authors could consider including other osteoblast regulators Alpl, Sp7 (Nakashima et al., 2002), and Htra1 (Iyyanar et al., 2019).

2.       Lines 45, 68 need references for the respective statements.

3.       Line 46, sentence is incomplete.

4.       The authors should use nomenclature suggested by Jackson laboratories for the nomenclature of mouse genes as italicized (eg., Runx2) and mouse proteins as capitalized (eg., RUNX2) and human genes as both italicized and caps (eg., RUNX2).

5.       Line 85, the authors state that the anterior-posterior axis of the heads was shorter. This statement cannot be made with skeletal preparations, as most of the soft tissues are digested at this stage during sample preparation. Anyhow, the lateral view of the head does not reflect the shortened head along the anterior-posterior axis.

6.       Figure legend 1, line 102, it must be dorsal view.

7.       Figure 2, the authors should make sure that the views are dorsal.

8.       Author should also explicitly mention how many Ns they confirmed the same finding for each figure panels in the results section.

9.       Numerical errors like for numbers less than 10, they have spelled out like “seven”.

Reviewer 3 Report

The manuscript by Jiang et al. present an impressive number of data regarding the tissue-specific contribution of the CBFB2 and RUNX2 in skeletogenesis. The authors provide very detailed analysis of the various mouse models used, and explored the possible connection and requirements for CBFB2 and RUNX2. The quality of the work is highly appreciated, and the methodological description very thorough. There are a few issues, however, that would need to be addressed before this manuscript can be considered for publication.

In the abstract and introduction, the use of the term 'haplodeficiency' should be revisited to haploinsufficiency.

It was unclear why there was so many red highlights throughout the manuscript.

Page 1, line 36: To avoid confusion, should Cbfb gene be named Cbfb1 instead? Or that would results in a complete absence of all CBFB proteins 1 and 2?

Page 2, line 47-48: The rescue experiment for the hematopoiesis is not clear. How was this done briefly? Please give reference.

Page 2, line 94: So the residual CBFB protein in the CBFB2-/- is CBFB1? Can the 2 CBFB isoforms be distinctly recognized on the western (different masses)?

Page 2, line 115-116: It seems that there are noticeable differences in the limbs of the Runx2+/- compared to WT. Especially for the scapula. Was any measurement done to substantiate the claim of no differences?

Page 4, line 127-128 (Figure 2): This may be just semantic, but the alizarin red staining in the indicated Cbfb2-/- tissues (scapula, sternum) appears to be the same as WT. Rather, the extent (surface area) of alizarin is slightly reduced. So the authors cannot really say that there is less 'mineralization' per se, but rather a decreased mineralized area.

Page 5, line 137: In this result section (2.3) about frontal suture, it is unclear why the authors focus more about the supplemental data than the actual data presented in figure 4 and 5. If they do not show more or different data, it could simply be said that 2 different ages were investigated with same results (or not) and then focusing on the differences. A general sentence at the beginning of the paragraph indicating that the parameters were measured at the 3-4 different ages would help.

Page 6, line 150-151: It would be important to note here that there are more col2+ cells and less Col10+ cells in the Cbfa2-/- mice.

Page 9, line 225 (Figure 6 legend): It should be indicated more precisely in the figure which portion of the rib was used for RNA extraction: cartilaginous, bony, or both segments. How do the authors explain the same levels of Runx2 expression detected here, while the protein levels shown in Figure 1 for RUNX2 were significantly ablated?

Page 10 (Figure 7): There is an inconsistency in the units for the clavicle measurements between the Cbfa2 (%) and Runx2 (cm2).

Page 18, line 375: How the authors explain the discrepancy between the lack of Runx2 and Runx3 mRNA measured by RT-qPCR and that of the protein contents?

Page 20, line 411 (Figure 14): The cortical regions shown in Figures 14 and 15 for the Runx2+/- seem to have increased porosity. It is not quite clear how the cells were counted. Therefore, the brdu index calculated and shown do not appear to be related to 'osteoblastic cells' (term used for the figure title) specifically, as there could be 'marrow' cell content within the cortices. It is impossible to know if the labelled cells are osteoprogenitors. Maybe a more general term should be used.

Author Response

Response to Reviewer 3

Thank you for the constructive suggestions.

The manuscript by Jiang et al. present an impressive number of data regarding the tissue-specific contribution of the CBFB2 and RUNX2 in skeletogenesis. The authors provide very detailed analysis of the various mouse models used, and explored the possible connection and requirements for CBFB2 and RUNX2. The quality of the work is highly appreciated, and the methodological description very thorough. There are a few issues, however, that would need to be addressed before this manuscript can be considered for publication.

In the abstract and introduction, the use of the term 'haplodeficiency' should be revisited to haploinsufficiency.

Thank you. We changed 'haplodeficiency' to ‘haploinsufficiency’.

It was unclear why there was so many red highlights throughout the manuscript.

The manuscript was a revised version. We had modified the manuscript of the first version according to the suggestions by two reviewers, and showed the changed parts red. These were returned to black.

Page 1, line 36: To avoid confusion, should Cbfb gene be named Cbfb1 instead? Or that would results in a complete absence of all CBFB proteins 1 and 2?

Page 1, Line 45 Cbfb-deficient (Cbfb–/–) mice lack all CBFB proteins (CBFB1 and CBFB2).

Page 2, line 47-48: The rescue experiment for the hematopoiesis is not clear. How was this done briefly? Please give reference.

We briefly described as follows:

Knock-in of green fluorescent protein (GFP) into the coding region of Cbfb maintained sufficient function in hematopoietic cells to bypass the early embryonic lethality (16). Further, the introduction of Cbfb into Cbfb–/– mice using Cbfb transgenic mice under the control of Tek1 or Gata1 promoter, which directs Cbfb to hematopoietic progenitor cells, rescued definitive hematopoiesis in Cbfb–/– mice (17, 18). These mice survived until birth but showed severely impaired intramembranous and endochondral ossification.

Page 2, line 94: So the residual CBFB protein in the CBFB2-/- is CBFB1? Can the 2 CBFB isoforms be distinctly recognized on the western (different masses)?

Yes. CBFB in Cbfb2–/– mice is CBFB1 in Fig. 1. CBFB1 is 187 amino acids and CBFB2 is 182 amino acids. We could not distinguish them in Western blot.

Page 2, line 115-116: It seems that there are noticeable differences in the limbs of the Runx2+/- compared to WT. Especially for the scapula. Was any measurement done to substantiate the claim of no differences?

We measured the total area and mineralized area of scapulae and the ratios in Cbfb+/+, Cbfb–/– Runx2+/+, and Runx2–/– mice. As the reviewer suggested, the total area and mineralized area of scapulae and the ratios were reduced in both Cbfb–/–and Runx2–/–mice compared with the respective control mice. We added the data in Fig. 2S and Fig. 3M and modified the results.

Page 4, line 127-128 (Figure 2): This may be just semantic, but the alizarin red staining in the indicated Cbfb2-/- tissues (scapula, sternum) appears to be the same as WT. Rather, the extent (surface area) of alizarin is slightly reduced. So the authors cannot really say that there is less 'mineralization' per se, but rather a decreased mineralized area.

Yes, alizarin red staining indicated a decreased mineralized area but not less mineralization as the reviewer pointed out. We described it as delayed mineralization.

Page 5, line 137: In this result section (2.3) about frontal suture, it is unclear why the authors focus more about the supplemental data than the actual data presented in figure 4 and 5. If they do not show more or different data, it could simply be said that 2 different ages were investigated with same results (or not) and then focusing on the differences. A general sentence at the beginning of the paragraph indicating that the parameters were measured at the 3-4 different ages would help.

In the first version, we showed the four figures (Figs. 4, 5 and Supplementary Figs. 2, 3 in the second version) in one figure, because the presentation of the phenotypes at four different ages is necessary to show the processes of the closure of posterior frontal suture. However, the reviewer pointed out that individual picture is too small to evaluate. The reviewer also suggested to enlarge and divide other figures. As we hesitated to increase the number of figures too much, we presented the two in the manuscript and the others in supplementary in the second version. Now, we presented four figures in the manuscript of the third version.

We also described as follows:

the processes were examined in Cbfb2–/– mice at four different ages

Page 6, line 150-151: It would be important to note here that there are more col2+ cells and less Col10+ cells in the Cbfa2-/- mice.

Thank you. We described as follows:

In Cbfb2–/– mice at P10, the frontal suture was still apparently open in the micro-CT image (Fig. 5C, D), frontal suture cells condensed, most of them expressed Col2a1, and a few expressed Col10a1 (Fig. 5G, H, K, L, O, P, S, T, W, X). Thus, there were more Col2a1+ cells and less Col10a1+ cells in Cbfb2–/– mice compared with Cbfb2+/+ mice, indicating that the process of endochondral ossification was delayed in Cbfb2–/– mice.

Page 9, line 225 (Figure 6 legend): It should be indicated more precisely in the figure which portion of the rib was used for RNA extraction: cartilaginous, bony, or both segments. How do the authors explain the same levels of Runx2 expression detected here, while the protein levels shown in Figure 1 for RUNX2 were significantly ablated?

The bony segments in ribs were used for RNA preparation.

Cbfb regulates the stabilization of Runx2 protein but not the transcription as we previously reported (J Bone Miner Res 30:706-14, 2015).

Page 10 (Figure 7): There is an inconsistency in the units for the clavicle measurements between the Cbfa2 (%) and Runx2 (cm2).

Thank you for the indication. It is a mistake. Both are mm. We corrected it.

Page 18, line 375: How the authors explain the discrepancy between the lack of Runx2 and Runx3 mRNA measured by RT-qPCR and that of the protein contents?

As CBFB stabilizes RUNX family proteins, including RUNX1, RUNX2, and RUNX3, by heterodimerizing with them, RUNX family proteins were reduced in Cbfb2–/– mice. However, CBFB has no ability to regulate the transcription of Runx family genes.

Page 20, line 411 (Figure 14): The cortical regions shown in Figures 14 and 15 for the Runx2+/- seem to have increased porosity. It is not quite clear how the cells were counted. Therefore, the brdu index calculated and shown do not appear to be related to 'osteoblastic cells' (term used for the figure title) specifically, as there could be 'marrow' cell content within the cortices. It is impossible to know if the labelled cells are osteoprogenitors. Maybe a more general term should be used.

As the cortical bone in Runx2+/- mice was detached in Fig. 16H (previous Fig. 14H), the section was replaced.

According to the suggestion, we changed “osteoblastic cells” to “osteoblast-like and osteoprogenitor-like cells”.

We counted the cells with the shape of osteoblasts or osteoprogenitor cells near the bone matrix and indicated these cells by arrows in new Figs. 16 and 17.  

Round 2

Reviewer 2 Report

appreciate the author for his revision.

authors response was accepted.

Author Response

Thank you for your review!

Reviewer 3 Report

All queries were adequately addressed. I recommend acepptance.